# Mucus Structure, Viscoelastic Properties, and Composition in Chronic Respiratory Diseases

**DOI:** 10.3390/ijms25031933

**Published:** 2024-02-05

**Authors:** Michela Abrami, Alice Biasin, Fabiana Tescione, Domenico Tierno, Barbara Dapas, Annalucia Carbone, Gabriele Grassi, Massimo Conese, Sante Di Gioia, Domenico Larobina, Mario Grassi

**Affiliations:** 1Department of Engineering and Architecture, University of Trieste, Via Valerio 6/A, I-34127 Trieste, Italy; michela.abrami@dia.units.it (M.A.); alice.biasin@phd.units.it (A.B.); mario.grassi@dia.units.it (M.G.); 2Institute of Polymers, Composites and Biomaterials, National Research Council of Italy, P.le E. Fermi 1, I-80055 Portici, Italy; fabiana.tescione@cnr.it (F.T.); domenico.larobina@cnr.it (D.L.); 3Clinical Department of Medical, Surgical and Health Sciences, Cattinara University Hospital, University of Trieste, Strada di Fiume 447, I-34149 Trieste, Italy; tiernodomenico@gmail.com (D.T.); ggrassi@units.it (G.G.); 4Department of Chemical and Pharmaceutical Sciences, University of Trieste, Via L. Giorgieri 1, I-34127 Trieste, Italy; bdapas@units.it; 5Department of Clinical and Experimental Medicine, University of Foggia, Via Napoli 121, I-71122 Foggia, Italy; annalucia.carbone@unifg.it (A.C.); sante.digioia@unifg.it (S.D.G.)

**Keywords:** asthma, chronic pulmonary obstructive disease, cystic fibrosis, mucus, low-field NMR, viscoelasticity, biomarkers

## Abstract

The respiratory mucus, a viscoelastic gel, effectuates a primary line of the airway defense when operated by the mucociliary clearance. In chronic respiratory diseases (CRDs), such as asthma, chronic obstructive pulmonary disease (COPD), and cystic fibrosis (CF), the mucus is overproduced and its solid content augments, changing its structure and viscoelastic properties and determining a derangement of essential defense mechanisms against opportunistic microbial (virus and bacteria) pathogens. This ensues in damaging of the airways, leading to a vicious cycle of obstruction and infection responsible for the harsh clinical evolution of these CRDs. Here, we review the essential features of normal and pathological mucus (i.e., sputum in CF, COPD, and asthma), i.e., mucin content, structure (mesh size), micro/macro-rheology, pH, and osmotic pressure, ending with the awareness that sputum biomarkers (mucins, inflammatory proteins and peptides, and metabolites) might serve to indicate acute exacerbation and response to therapies. There are some indications that old and novel treatments may change the structure, viscoelastic properties, and biomarker content of sputum; however, a wealth of work is still needed to embrace these measures as correlates of disease severity in association with (or even as substitutes of) pulmonary functional tests.

## 1. Introduction

Chronic respiratory diseases (CRDs) represent a group of pathologies that in 2019 were the third leading cause of death and responsible for 4.0 million deaths [1]. Some of the most common are asthma and chronic obstructive pulmonary disease (COPD), albeit these non-communicable diseases include rare genetically determined cystic fibrosis (CF).

COPD is the third leading cause of death worldwide, causing 3.23 million deaths in 2019. Tobacco smoking accounts for over 70% of COPD cases in high-income countries, whereas in low- and middle-income countries tobacco smoking accounts for 30–40% of COPD cases, and household air pollution is a major risk factor [2].

Asthma affected an estimated 262 million people in 2019 [3] and caused 455,000 deaths worldwide [4]. Many factors have been linked to an increased risk of developing asthma, including familiarity, urbanization, events in early life (low birth weight, prematurity, exposure to tobacco smoke and other sources of air pollution, as well as viral respiratory infections), overweight or obesity, and exposure to a range of environmental allergens and irritants (indoor and outdoor air pollution, house dust mites, molds, and occupational exposure to chemicals, fumes, or dust) [4].

CF is an autosomal recessive disease due to mutations in the *CF Transmembrane Conductance Regulator* (*CFTR*) gene, which in Caucasian populations has an incidence between 1/3000 and 1/6000, corresponding to carrier rates of 1/28 and 1/40, respectively. Due to progress in disease screening and better targeted therapies, the estimated median age of survival of CF patients today is close to 50 years [5].

Being incurable, COPD, asthma, and CF impose a very high socioeconomic burden due to the continuous cures given to these patients all their life and the chronicity of these diseases. For example, it has been recently estimated that COPD will cost the global economy INT$ 4326 trillion in 2020–2050 [6].

In this review, we wish to survey the role of the mucus structure and viscoelastic features in CRDs in trying to explore whether these characteristics and properties could advance our knowledge not only of the pathophysiology of these non-communicable diseases but also their usefulness as biomarkers of prognosis and therapeutic efficacy. We anticipate that in comparison with classic sputum inflammatory mediators and metabolites, much work still needs to be done in considering mucus structure and viscoelasticity as biomarkers of CRD pathologies. This study was conceived as a narrative review and not a systematic one; thus, we have surveyed the most significant and important papers in the fields of healthy and pathological mucus and CRDs by performing an electronic search in the PubMed, EMBASE, and Scopus databases.

### 1.1. Respiratory Mucus in Healthy and Diseased Conditions

Despite strong evidence that deviations in mucus abundance and/or composition drive mortality and morbidity in several airway diseases, measuring mucus and mucins remains a challenge. Obtaining mucus in vivo from the trachea, bronchi, and bronchioles can be difficult due to limitations in collection methods. On the other hand, animal models used to study respiratory diseases [7] show marked differences in their airway anatomy and structure, including abundance of submucosal glands. Nevertheless, animal species with similar airway architectures and physiologies, such as pigs, have allowed researchers to identify some mucus properties in health and disease [8], which will be recalled later on in the following sections. The use of in vitro systems as alternatives to animal models might not be an accurate representation of the in vivo environment [9]. Finally, sputum also provides information about mucus and mucins. This heterogeneous material is expelled from the lower airways via coughing. There are two major types of sputum: induced and spontaneous. An advantage of spontaneous sputum is that no clinical intervention is required for its production. Conversely, induced sputum entails aerosolization of hypertonic saline to the airways using standardized protocols. Limitations of both spontaneous and induced sputum include the possibility for saliva contamination, as well as variations in the amount of sputum produced. Despite these considerations, sputum is widely used to study airway mucus and airway inflammation [10].

Human airway mucus was defined by analyzing mucus from individuals without respiratory disease. Macroscopically, mucus is a non-Newtonian, thixotropic gel; under low shear, mucus behaves like an elastic solid, and under high shear, mucus behaves like a viscous liquid and eventually deforms irreversibly [11]. In general, at low shear rate, mucus viscosity is around 10^3^ Pa·s, but at a shear rate near the physiological maximum (achieved during coughing), the viscous drag is greatly reduced to 10^−2^ Pa·s.

In all organs, the viscoelastic properties of mucus are closely regulated biochemically; this regulation has to be efficient to ensure (1) efficient clearance through cilia movement and (2) maintenance of sufficient adhesive and elastic strength to be retained on the epithelial surface for its toxins barrier action [12]. Mucus consists of mucins, DNA, lipids, ions, proteins, cells and cellular debris, and water. A biochemical dysregulation of these various constituents could adversely affect physical mucus properties, contributing dramatically to disease conditions. Rheological properties are mainly affected by the mucus hydration degree and mucin content, which usually range, respectively, between 90 and 98% and 2 and 5% by weight for healthy lung mucus [13]. The presence of DNA in mucus originates from debris of shed epithelial cells and represents 0.02% [14]. Studies on purulent CF sputum individualized the main contribution of neutrophil-derived DNA on the consistency and nature of those thick and viscous mucus samples [15]. Rheological properties could be well-represented by yield stress, γ_c_; basically, when the applied shear stress on the material is higher than γ_c_ of the material, the material exhibits liquid-like behavior. If the applied shear stress is lower than γ_c_, the material shows solid-like properties. For healthy mucus, γ_c_ falls below 1 Pa·s. In contrast to healthy mucus, γ_c_ for CF sputum ranges from 0.1 to 100 Pa·s, while γ_c_ for COPD and asthmatic sputum ranges between 1 and 40 Pa·s.

To further characterize the properties of normal/pathological mucus, it is possible to study the movement of a nano-metric particle within the mucus itself. At this purpose, Schuster and co-workers [16] focused their attention on nanoparticle transport inside fresh human mucus samples by means of high-resolution fluorescence microscopy and multiple-particle tracking analysis (MPT). In order to minimize nanoparticles’ muco-adhesion, an event that would have altered the output of MPT analysis, they coated carboxylate-modified polystyrene nano-spheres (PS-COOH) with low-molecular-weight polyethylene glycol (PEG). In addition, in order to easily explore small and large meshes, they used nanoparticles sized 100, 200, and 500 nm in diameter. They found that in un-coated conditions, even the movement of smaller particles was hindered. On the contrary, PEG coating allowed 100 (84%) and 200 (75%) nm particles to diffuse almost freely. The 500 nm particles showed constrained traces indistinguishable from both conditions. Thus, they concluded that airway mucus is a selectively permeable barrier where particle size and adhesiveness determine transport properties.

### 1.2. Clinical and Pathological Aspects in CRDs

COPD, asthma, and CF are all characterized by overproduction of an abnormal respiratory mucus, which in turn may cause the main sign and symptoms affecting these patients. In COPD, symptoms include chronic cough, sometimes with thick mucus (sputum), difficulty breathing, wheezing, and tiredness. Common symptoms in asthma include a persistent cough, especially at night, wheezing at exhalation, shortness of breath or difficulty breathing, and chest tightness. CF patients experience persistent cough that produces sputum, wheezing, exercise intolerance, inflamed nasal passages, and recurrent sinusitis.

Besides sputum production, these three CRDs show quite identifiable pathological pictures in the respiratory tract. COPD is characterized by poorly reversible airflow obstruction and an abnormal inflammatory response in the lungs, resulting in mucous hypersecretion (chronic bronchitis), tissue destruction (emphysema), and disruption of normal repair and defense mechanisms causing small airway inflammation and fibrosis (bronchiolitis) [17]. On a chronic course, COPD patients experience acute exacerbations, which are mostly induced by bacterial and viral infections [18,19,20].

In asthma, the airflow obstruction is reversible; however, the bronchioli are burdened over time by the inflammatory process, leading to inflammatory cell infiltration and thickening of the airway wall with increased smooth muscle mass [21].

Patients with CF have a robust inflammatory response in the airways, mucopurulent plugging of small- and medium-size bronchioles, and bronchiectasis [22]. CF airways are prone to opportunistic bacterial infections, including those caused by *Pseudomonas aeruginosa*, *Staphylococcus aureus*, the *Burkholderia cepacia* complex, *Achromobacter* species, and *Haemophilus influenzae* [23].

### 1.3. Pathophysiologal Changes in CRD Mucus/Sputum

The airway mucus, although composed mostly of water (95%), has many physical characteristics of a solid. The 2–3% of the solid phase is composed of mucins, non-mucin proteins, enzymes, salts, lipids, and cellular debris, which allows the mucus to be classified as a gel with properties of both a soft, elastic solid and a viscous fluid [24]. Mucins, exceedingly large glycoproteins (up to 0.5 × 10^6^ D per monomer) [25], are the main solid component and are present in airway mucus in different families [26]. Secreted, gel-forming mucins (predominantly MUC5AC and MUC5B, with smaller contributions from MUC2, MUC8, and MUC19) are those more expressed in the airways. In healthy conditions, while MUC5AC and MUC5B are the main gel-forming mucins produced by goblet cells, submucosal glands secrete only MUC5B [24,27,28,29].

#### 1.3.1. Physiology of Respiratory Mucus Secretion

Epithelial cells covering various surfaces throughout the body employ specialized mechanisms to establish a robust barrier. Within the intrapulmonary airways, surface epithelial cells and submucosal glands release a non-adherent layer of mucus. This mucus is subsequently moved from distal to proximal airways and expelled from the lungs through ciliary beating, serving to eliminate inhaled particles, pathogens, and toxicants [30]. In general, mucus is characterized by two different layers: the gel layer and the periciliary liquid layer (PCL). While the first can block the penetration of invading pathogens, the other contributes to ciliary motion and mucociliary clearance [31]. The surface epithelium close to the mucus layer is composed of two major cell types: ciliated cells and secretory cells (goblet cells and club cells). The secretory granules into *goblet cells* are very important because they allow for the storage and secretion of mucins, while ciliated cells are involved in the transportation of secreted mucus [32]. Besides mucins, respiratory epithelial cells secrete a large number of non-mucin proteins, including defensins, enzymes, antibacterial peptides, immunoglobulins, protease inhibitors, and oxidants [33]. Granules containing mucus are exocytosed and swiftly hydrated. In goblet cells, granule exocytosis occurs either under basal conditions at a low level or regulated by extracellular stimuli. The mucins and other components are packed in cytosolic-stored granules, which are then moved to the cell apical surface. The myristoylated alanine-rich C-kinase substrate (MARCKS) protein is essential in this movement. Phosphorylation of MARCKS allows it to bind to actin/myosin fibers, which contracts and encourages the granules to fuse with the plasma membrane. Soluble N-ethyl-maleimide-sensitive factor attachment protein receptor (SNARE) proteins are also involved in tethering granule membranes with plasma membrane before exocytosis can occur [34]. Upon exocytosis and dilution of Ca^2+^ present in granules as a shielding cation, mucin expansion occurs due to electrostatic expulsion of polyanionic mucin glycoproteins. Various gases (e.g., cigarette smoke), neurohumoral factors (e.g., cholinergic agonists), and inflammatory mediators (e.g., cytokines) can upregulate basal granule exocytosis and increase MUC5AC expression [34].

In the respiratory tract characterized by a diameter greater than 2 mm, the process of mucus secretion is mainly due to *submucosal glands*. Within the main bronchi of humans, the glandular volume is approximately 50 times greater than that of surface goblet cells; however, as one moves distally, the glands decrease in both size and occurrence [35]. Each gland is connected to the lumen of the respiratory tract through a duct, the more superficial part of which is characterized by the presence of ciliated cells that enable the movement of mucus toward the respiratory tract lumen [24]. Goblet cells and serous cells, lining the acinar region of the submucosal gland, release a diverse range of innate immune proteins, mucins, and electrolytes. These components play a crucial role in regulating the hydration of the secretions. Airway submucosal glands can undergo swift and substantial secretory responses triggered by irritants and neural stimulation, cyclic AMP, purinoreceptor and muscarinic activation, and substance P [35]. Submucosal glands release mucins, with MUC5B being the primary component as well as smaller quantities of MUC5AC. Additionally, these glands secrete a diverse array of innate host defense proteins and antimicrobial peptides, including lysozyme, lactoferrin, β-defensins, and surfactant proteins SP-D and SP-A, among others [36]. The proteinaceous product from submucosal glands are intricately regulated in conjunction with fluid and electrolyte transport, facilitating the prompt release and spread of mucus onto the surfaces of the airways. This process is essential for the movement of the mucus gel upward along the airway through ciliary activity. Serous cells actively secrete chloride (Cl^−^) and bicarbonate (HCO_3_^−^), contributing predominantly to the fluid composition of glandular secretions. Glands receive abundant innervation from tonically active, reciprocally stimulating intrinsic neurons within the airways. The majority of glandular mucus is consistently secreted in vivo, with substantial, temporary surges induced by the emergency reflex stimulation from the vagus. Increases in intracellular concentrations of cAMP and Ca^2+^ synchronize the secretion of electrolytes and macromolecules, likely concurrently manifesting as baseline activity in vivo. The cholinergic elevation of Ca^2+^ appears primarily responsible for transient enhancements in secretion [35].

Mucus constitutes an intricate blend of mucins and a variety of antimicrobial proteins, metabolites, fluids, and electrolytes. Its quantity and composition exhibit variations along the cephalocaudal axis of the lung and are influenced by environmental exposures and inflammation. Airway *mucins* constitute a group of polymeric, extensively glycosylated proteins generated by both airway epithelial cells and submucosal glands. There are a minimum of 21 genes that encode human mucins, with 16 of them being expressed in airway epithelial cells [35,36,37]. Mucins can be categorized into (1) non-polymerizing, secreted mucins; (2) cell-associated mucins anchored at cell surfaces; and (3) gel-forming mucins, with MUC5B and MUC5AC being the most abundant among them.

Under normal conditions, both club cells and goblet cells in the airways produce and secrete mucins. During hydration, mucins experience rapid expansion and undergo quaternary structural changes as the layers of mucociliary gel are formed. Fluids, along with secreted proteins, electrolytes, calcium, and metabolites, interact with the mucus, establishing the periciliary fluid layer. This layer serves as the platform on which the gel-forming mucins, including MUC5B and MUC5AC, along with their associated cargoes, ascend the airways through ciliary activity [38].

In humans and pigs, the upper airways are endowed with submucosal glands, the main site for the production of MUC5B. Once the packed mucin (MUC5B) is assembled into granules, Cl^−^ and HCO_3_^−^ secretion-induced flow generates a force that pulls out and helps to unfold the mucin threads. The threads gather and become thicker, and in the gland opening, thick mucin forms bundles [39]. The bundles’ core, made up of MUC5B mucin, is further covered by MUC5AC mucin [29], which is secreted from the goblet cells located in the distal ducts of the glands and the epithelial surface. The MUC5AC coating potentially offers anchoring activity that retards the movement of the mucus gel as periciliary fluids move upward along the airway [29,36].

There are a number of questions that are still unresolved in the field of mucus synthesis and secretion that would guarantee a better comprehension of mucus physiology and pathophysiology if addressed. For instance, single cell RNAseq data have revealed a novel cell type in the airways of humans and mice, named “ionocyte”, which expresses high levels of CFTR and V-type ATPases (as well as ENaC). This cell type looks to be prevalent in the proximal airways, particularly in the ductal regions of submucosal glands. The role of ionocytes in mucus secretion and mucociliary clearance is still an open question [40,41,42], but they were hypothesized to be involved in the production of a mildly hypotonic submucosal gland secretion [43]. On the other hand, the hydration of airway surface fluid (ASL), i.e., the liquid in the mucus layer and PCL compartments, is finely tuned by ion secretion/absorption through CFTR and ENaC; however, recent research is trying to uncover the role of other ion channels, such as the interrelationship between the calcium-activated channel TMEM16a and intracellular Ca^2+^ signals in club cells and goblet cells [43,44,45].

#### 1.3.2. Mucins, Goblet Cells, and Submucosal Glands in Pathophysiology of Mucus Secretion

Sputum production is the hallmark of CRDs, with its own peculiarities depending on the disease, although COPD, asthma, and CF show increased mucus amounts localized in the airways (Table 1). Both asthmatic and bronchitis respiratory secretions contained more MUC5AC mucin compared to those collected from normal individuals [46]. While the increase in MUC5AC expression in asthma is consistent, results concerning MUC5B are more conflicting [36,47,48,49]. In COPD airways, MUC5AC and MUC5B expression is much increased, and their expression patterns are altered compared with those of smokers and normal subjects [47]. Kirkham et al. found that MUC5AC was the predominant mucin in the sputum of smokers, whereas MUC5B, predominantly the lower-charged, glycosylated form, was more abundant in the sputum from COPD patients [50]. Histopathological analyses revealed that COPD is associated with increased expression of MUC5B in the bronchiolar lumen and of MUC5AC in the bronchiolar epithelium [51]. Furthermore, it was shown that MUC5AC expression is increased in the bronchial airways, and particularly in the submucosal glands, of COPD patients compared with both smokers with normal lung function and non-smokers [52].

Although in the beginning a reduction in mucin expression in CF airways was observed [53,54], when the analysis was performed using physical rather than immunological methods, MUC5AC and MUC5B were found to be increased in comparison to non-CF control samples independently of whether they were detected either in sputum or bronchoalveolar lavage fluid (BALF) [55,56]. This discrepancy can be explained by mucin proteolysis in the harsh CF luminal airway content and thus loss of immunological reactive epitopes. CF sputum shows increased viscoelasticity due to mucus hyperconcentration and increased osmotic pressure caused by dehydration [43,57] and oxidative stress-mediated mucin disulfide cross-links [58,59], which correlates with the severity of lung disease [60,61]. The CF pig model has been instrumental in defining some abnormalities in mucin secretion and biophysical properties in pathology. Immunohistochemical studies have revealed that while MUC5B more often filled CF submucosal gland ducts, MUC5AC sheets accumulated in the airways overlying MUC5B strands [28]. Studies in CF piglets have shown that increased osmotic pressure and cohesive forces reflected hyperconcentration of mucus secreted by submucosal glands and the airway surface epithelium, shedding light on disease-initiating mucus accumulation in CF lungs [62]. In the CF pig model, it has also been observed that mucus strands remained tethered to gland ducts [63] or trapped on the tracheal surface attached to the surface goblet cells [63,64]. Finally, in humans, bundles of F-actin and DNA released from dead inflammatory cells and superimposed to the mucus mesh contribute to the altered viscoelastic properties of CF sputum, thus inhibiting clearance of infected airway fluid [58,65,66].

Sputum from patients with asthma is more viscous than that from patients with chronic bronchitis or bronchiectasis [67,68]. Increased viscosity of mucus plugs in asthma can be the result of a high concentration of mucins, non-covalent interactions between extremely large mucins assembled from conventional-size subunits, and their low charged density [69]. Airway plasma exudation is a distinguishing feature compared with COPD or CF and may contribute to the hyperviscosity of asthmatic sputum [70]. In addition, intraluminal mucus in asthma is in continuity or adheres to goblet cells, a peculiarity not present in chronic bronchitis, leading to “tethering” of luminal mucins to the airway epithelium [71]. The release of mucins in chronic bronchitis airways might be due to elastase produced by neutrophils, the predominant inflammatory cell in COPD [72], while in asthma, predominant eosinophils do not generate the appropriate proteases to facilitate mucin release [73].

On the other hand, Selisier and colleagues assessed that mucoid COPD samples demonstrated significantly greater viscoelasticity than sputum from both CF and normal subjects [74]. Indeed, the viscosity of CF secretions appears to be no greater than that of sputum from patients with bronchiectasis or chronic bronchitis [68,75,76]. The obstructed airflow in CF could then be more the result of a mucociliary clearance defect than mucus hyperviscosity.

Mucus overproduction not only depends on MUC5AC/B overexpression, and other contributors, like goblet cell hyperplasia and submucosal gland hypertrophy, should be mentioned. In inflammatory airway diseases, mucus hypersecretion from metaplastic and hyperplastic goblet cells leads to mucus trapping and airflow obstruction of airways, especially in the smallest conducting airways [77]. Another hallmark of mucus overproduction in CRDs is the enlargement of the bronchial glands in the cartilaginous airways (larger than 2 mm in diameter) [78]. Thus, in COPD, submucosal gland hypertrophy is observed in the large airways [78,79], and the excessive luminal mucus correlates with the amount of glands [80], although no correlation has been found between mucous gland enlargement and sputum production [81,82]. Goblet cell hyperplasia is another feature of both large and small airways in chronic bronchitis [83].

In asthma, airway remodeling involves goblet cell hyperplasia along with thickening of the basement membranes and hyperplasia/hypertrophy of smooth muscle cells [84]. Submucosal gland hypertrophy is also observed in asthma [85,86], with notable changes compared with glands in chronic bronchitis; in asthma, although two to four times greater in size compared to controls [87], the glands appear morphologically normal with an even distribution of mucous and serous cells, while in chronic bronchitis, gland hypertrophy is characterized by an increased number of mucous cells relative to serous cells [88]. It is worth noting that both goblet cell hyperplasia and glandular hypertrophy are not common to all patients, indicating that other factors contribute to mucus hyperproduction, namely immuno-inflammatory responses (see below Section 1.4) [86]. Moreover, the contribution of mucus hypersecretion and mucous metaplasia to airway flow obstruction remains to be determined compared with other contributors, such as extravasated plasma, smooth muscle contraction, and wall thickening [89].

The involvement of submucosal gland hypertrophy has been identified since the first studies identified their hyperplasia and mucin occlusion of the gland ducts as early histological hallmarks of CF (reviewed in [90]). However, later on, from the identification of the CFTR dysfunction as the primeval cause of CF, the hypothesis of the uncoupling of mucus and liquid secretion at the glandular level as the main cause of mucus plug formation took place and is now included at the onset of CF lung disease [43,90,91,92]. Because submucosal glands, in their serous cell components, secrete antimicrobial proteins and peptides that prevent infection by killing bacteria and inhibiting viruses [35,93,94,95,96,97], impairment in fluid secretion in CF airways may also be responsible for opportunistic infections aggravating the lung disease. It is worth noting that the acidic pH of CF sputum is implicated more in the deficiency of the airway antimicrobial defense than alterations of viscoelasticity and hence in decreased mucociliary clearance [59,98].

### 1.4. Pathophysiology of Mucus Production in CRDs

A plethora of biochemical mediators and signaling pathways are involved in the pathophysiology of CRDs. Here, we will focus on those related to mucus hyperproduction and altered mucin expression (Figure 1).

In CF, mutations in the *CFTR* gene are the primary engine causing mucin hypersecretion and concentration in the airway surface liquid [99,100]. Many factors may contribute, and some of them can be attributed directly to CFTR lack/dysfunction, such as dehydration and acidic pH due to aberrant Cl^−^ and HCO_3_^−^ secretion and dysregulation of epithelial sodium channel (ENaC)-mediated Na^+^ transport resulting in ASL water hyperabsorption and reduction in mucus mesh size (e.g., entanglement). On the other hand, impaired Ca^2+^ granule sequestration prevents mucin unpacking and expansion (e.g., compaction). Other factors can derive from neutrophilic inflammation such that oxidative stress can introduce additional ionic, hydrogen, hydrophobic, and disulfide bonds and exaggerate entanglement and compaction of mucins [99]. Up-regulation of MUC gene expression, epithelial remodeling by goblet cell hyperplasia/metaplasia, and/or glandular hyperplasia are other mechanisms invoked to explain mucus hypersecretion in CF [101]. Recently, studies in newborn pigs showed a similarity between CF and non-CF submucosal glands with respect to cell types and transcript levels [102], signifying that it is the loss of ion secretion rather than an intrinsic cell defect that determines an increase in the elasticity and tensile strength of mucus strands, thereby preventing normal mucociliary transport [103]. Moreover, the defect of Cl^−^ and HCO_3_^−^ secretion by CF submucosal glands contributes to the pH acidification occurring in the ASL, the decrease in liquid secretion, and the increase in protein concentration, functions mediated directly by CFTR anion channels, leading to the loss of activity of antimicrobial peptides and proteins and mucociliary clearance disruption [63,97,102]. On the other hand, neutrophil elastase (NE) and defensin HNP-1 increase MUC5AC at both transcriptional and posttranscriptional levels in airway epithelial cells [104,105,106,107]. Neutrophil elastase also has a role in mucus cell metaplasia [108] and as a mucin secretagogue [109]. IL-17, a family of cytokines recruiting neutrophils and found at elevated levels in CF airways [110,111], increases MUC5AC [112,113] and MUC5B [113,114] gene expression in primary differentiated human bronchial epithelial (HBE) cells through activation of NF-κB (Figure 1). In vitro data have pointed to the role of dehydrated mucus in eliciting lumen macrophages and hypoxic airway epithelial cell secretion of IL-1β and IL-1α, the most potent segretatogues present in the mucopurulent sputum [43,57,115].

Due to similarities in the clinical phenotype between CF and COPD, it was hypothesized that mucus hypersecretion in COPD could be due to CFTR dysfunction. Indeed, smokers with and without COPD have reduced chloride conductance in lower airways, and this defect is associated with the presence of chronic bronchitis and the severity of dyspnea [116]. Cigarette smoke, via radical oxygen species (ROS) and cadmium/acrolein content, can induce CFTR dysfunction at the level of gene transcription, mRNA stability, protein stability, and protein function [117] (Figure 1). CFTR dysfunction leads to the production of highly viscoelastic and acidic secretions, which in turn contribute to mucus obstruction and potential plugging [118]. Cigarette smoke also increases mucin MUC5AC synthesis via epidermal growth factor receptor (EGFR) activation in the airway epithelial cells in vitro and in vivo [119,120] (Figure 1). Another study suggested that basic fibroblast growth factor, likely produced by inflammatory cells, e.g., neutrophils, in submucosal glands [121], may have a role in promoting mucus hypersecretion in smokers. Neutrophils play a key role in goblet cell stimulation and mucus hyperproduction in COPD via numerous mechanisms: (i) secretion of tumor necrosis factor (TNF)-α, which induces EGFR expression in airway epithelial cells; (ii) release of reactive oxygen species (ROS), which activate EGFR; (iii) activated NE cleaves the EGFR proligand, pro-transforming growth factor (TGF)-α, releasing TGF-α, which activates EGFR in a ligand-dependent fashion; and (iv) NE causes potent goblet cell degranulation [122]. Numerous other mediators could be implicated in COPD hypersecretion and goblet cell hyperplasia, namely IL-13, IL-1β, IL-4, CXCL8/IL-8, bacterial endotoxin lipopolysaccharide (LPS), matrix metalloproteinase (MMP)-9, etc., and they are under scrutiny to find effective drugs that are still in the experimental research stage [83,123].

Abnormal cell differentiation and mucus overproduction are determined in asthma through numerous mediators [36,124]. EGFR, which is increased in the asthmatic epithelium [125,126], is a driver of abnormal cell differentiation and epithelial dysfunction [127,128]. Downstream of the EGFR signals or TNF-α, several transcription factors are known to be involved in expression of the *MUC5AC* gene [129] in airway epithelial cells, particularly in goblet cells [101,129,130]. Disruption of epithelial barrier integrity results in activation of the epithelial cells and the release of various pro-inflammatory mediators (e.g., eicosanoids) and alarmins (such as uric acid, ATP, IL-1α, high mobility group box 1 (HMGB1), S100 proteins, and Hsp90α), triggering type two immunity and dysregulation of mucus production [131]. Type two immune cells, including type two T helper (Th2) cells and innate lymphoid cells (ILC2s), which orchestrate allergic airway inflammation, produce IL-13, which induces inhibition of ciliated cell differentiation and increased differentiation of goblet cells and goblet cell hyperplasia, producing high levels of MUC5AC [128,132,133,134] (Figure 1). A model according to which EGFR and IL-13 cooperate in determining goblet cell transition as well as mucus production has been envisioned [87]. IL-4, another Th2-dominant cytokine in asthma, sharing a promiscuous receptor with IL-13, has overlapping effects on goblet cell hyperplasia; however, IL-13 dominates over the other one [129].

## 2. Mucus Structure in CRDs

Organs exposed to the external environment are covered by a gel-like substance that acts as a first barrier against pathogens, named mucus. It is secreted by specialized goblet cells in the columnar epithelium, which cover the respiratory tract, the gastrointestinal tract, the reproductive tract, and the oculo-rhino-otolaryngeal tracts [135]. Mucus plays a crucial role in several other functions, such as the maintenance of epithelium hydration, the lubrication of tracts for particle passage, and its function as a barrier for noxious substances while being permeable for gas and nutrients [136].

Mucus could be named hydrogel due to its primary composition of water (95%) and macromolecular glycoproteins known as mucins (2–3%), with smaller components of proteoglycans (0.1–0.5%) and lipids (0.3–0.5%). Among these compounds, the most relevant for the mucus viscoelastic properties are mucins [137]. Mucins are large extracellular glycoproteins characterized by a heavy glycosylation (75–90%) that significantly contributes to the remarkable molecular weight ranging from 0.5 to 20 MDa. Mature mucins fall into two broad classes: the membrane-bound and the secreted mucins [138]. While these two classes share many common features, most of the secreted mucins are much larger than the membrane-bound mucins and more significantly contribute to the viscoelastic properties of the extracellular mucous layer [137]. Secreted mucins contain cysteine-rich domains located at both the amino and carboxyl termini. These domains are linked covalently via disulfide bonds to form mucin dimers that, after heavily glycosylation, further multimerize to form long, linear oligomers with adhesive and space-occupying properties [139].

The walls of the respiratory tract are covered by a thin aqueous layer, the periciliary layer (*PCL*), which has a thickness of about 5–10 μm in healthy subjects and patients [140] (see Figure 2). Inside the *PCL*, cells cilia beat with a typical frequency of 20 Hz and an amplitude of about 5 μm. The *PCL* is coated by another layer, named mucus [141], which has a thickness of about 25–30 μm in healthy subjects and CF patients [141,142], but it can reach 300 μm in the case of COPD patients [143]. Together, the *PCL* and the mucus layer (*ML*) constitute the airway surface layer (*ASL*) [144]. While the *ML* traps inhaled particles and transports them out of the lung through cilia-generated forces, the *PCL* provides a favorable environment for ciliary beating and cell surface lubrication [145]. Because CF, COPD, and asthma patients suffer from ASL dehydration, which implies the enormous increase in the concentration of many substances, such as mucin and globular proteins, mucus clearance is highly reduced, and *ASL* becomes the ideal nest for different types of pathogens.

While it is well known that the *ML* shows viscoelastic gel-like properties, the *ASL* “gel-on-brush” model [145] points out that the *PCL* is not a “simple” liquid phase, as commonly believed. Indeed, the *PCL* hosts membrane-spanning mucins and large muco-polysaccharides that are tethered to cilia, microvilli, and the epithelial surface. All of these polymers form a three-dimensional network impeding the capacity of *ML* mucins and inhaled particles to penetrate the *PCL* environment. The relatively high concentration of membrane-tethered macromolecules in the *PCL* produces intermolecular repulsion within this layer, which stabilizes the *PCL* against compression by an osmotically active mucus layer. In addition, the mutual repulsion of the tethered macromolecules gives origin to an osmotic pressure that regulates the hydration of the *PCL*. When the *ML* osmotic pressure exceeds that of the *PCL*, the *ML* compresses the *PCL,* slowing down or almost stopping cilia action and, consequently, mucus clearance.

The respiratory tract consists of two types of cells: ciliated and secretory cells. The *PCL* allows the cilia to beat without catching in the mucus and acts as a lubricant that allows the mucus to slide along the interface [146]. Secretory cells are further divided into Clara cells, goblet cells, and submucosal cells [147,148]. Mucin secretion occurs through regulated processes, and it is rapidly induced by a variety of secretagogues, like cholinergic and purinergic agonist, secreted inflammatory cell products, and pathogens [149]. In particular, MUC5AC and MUC5B are gel-forming mucins that constitute the most abundantly secreted mucins (75%) in the intrapulmonary airways and lend mucus its rheological properties [57]. MUC5AC is mostly produced in the proximal airways by goblet cells, while MUC5B is produced by submucosal glands throughout the airway [150]. Mucins are produced both in normal and stress conditions; however, their production significantly increases in the course of respiratory diseases. Whereas cilia are regular in size and the PCL has a constant thickness (≈7 μm) [24] in the respiratory tract, the mucus thickness varies significantly. In particular, it increases in thickness from the terminal bronchioles (0.1 μm) to the trachea (100 μm) [151]. When mucin production is increased, mucus becomes hyperconcentrated, much thicker, and more difficult to clear, thereby affecting mucociliary clearance. These events are shared by nearly all obstructive pulmonary diseases, i.e., CF, COPD [152], and asthma [153].

### 2.1. Mucus in CF

Batson and coworkers [154] used a size exclusion chromatography coupled to laser photometry to estimate the concentration of mucins in CF baseline patients and in those with exacerbations. Both stable and acute CF subjects presented an approximately fourfold increase of mucin concentration compared with control healthy subjects. A hyperconcentrated mucus hinders the clearing activity promoted by cough, favoring mucus stasis and bacterial infection. Indeed, Button et al. demonstrated that the increase in sample viscosity and elasticity produced slower airflow-mediated transport [155]. Coughing can move mucus from the lung through an adhesive failure, i.e., breaking mucus–cell surface adhesive bonds, and/or through cohesive failure, i.e., directly fracturing the mucus. These studies were performed using in vitro peel-test systems to directly measure the adhesive and cohesive forces of mucus produced by human bronchial epithelial (HBE) cultures from normal/CF/COPD patients. The magnitude of adhesive strength was measured with peel-tests at several velocities showing in the hyperconcentrated CF mucus, and higher force was required to tear at all velocities compared with mucus that was normally concentrated. These findings underline the crucial role of the hyperconcentrated mucus in pulmonary obstructive disease. In one of our previous works [156], we analyzed using Low-Field Nucleic Magnetic Resonance (*LF-NMR*) the sputum (derived from the lung mucus through expectoration) of CF and healthy patients. *LF-NMR* relies on the ability of hydrogen atom dipole (μ) to react to the perturbation of an external constant magnetic field *B*_0_, where hydrogen atoms are embedded (Figure 3). Basically, after a temporary *B*_0_ perturbation obtained through the application of a radio frequency pulse *B*_1_ perpendicular to *B*_0_, the dipoles tend to return to their initial alignment with *B*_0_ (relaxation). The relaxation rate is expressed by the inverse of a characteristic time constant *T*_2_ called spin–spin or transverse relaxation time [157]. This relaxation process is inversely proportional to the solid component inside sputum. Notably, the pathological increase in different substances, such as proteins, biological polymers, and mucin, in CF/COPD is well known [158]. Hence, we found a correlation between T_2_ and the amount of such elements in the sputum of CF patients [158]; moreover, we observed an inverse correlation with the systemic inflammatory marker CRP and the local inflammatory markers, such as IL-1β and TNF-α [156]. In subsequent work [152], we studied CF sputum characteristics through the combined use of *LF-NMR* and rheology. In particular, we investigated the correlation of *T*_2_ with (1) sputum viscoelasticity, (2) the mucociliary clearability index (*MCI*)/cough clearability index (*CCI*), and (3) the sputum average mesh size [154]. Rheology is used to describe and assess the flow behavior of materials. Fluids flow at different speeds, and solids can be deformed to a certain extent.

Depending on their physical behavior, materials can be put into liquids on one side and solids on the other. Rheological measurements, including viscosity (resistance to flow) and elasticity (stiffness), are often used together to describe the consistency of mucus. The rheological properties of mucus vary as a function of shear stress, time scale (rate) of shearing, and length scale. Changes in the rheological properties of mucus may greatly affect its ability to function as a lubricant, selective barrier, and the body’s first line of defense against infection. In addition, rheological characterization can provide insight into the nanostructure of a gel-like material, such as mucus. Mucus rheological analysis performed at different frequencies allows for determining *MCI* and *CCI* indices. We found an inverse correlation between elastic and viscous properties versus *T*_2_, indicating that a worsening of the lung condition (as evaluated by the test named “forced expiratory volume in the first second”, FEV_1_) is paralleled by an increase in mucus viscoelasticity, thus favoring stasis and the consequent inflammation. A direct correlation was also observed between *T*_2_ and *MCI*/*CCI*, showing that *T*_2_ provides information regarding airway mucus clearing. Moreover, there was a direct correlation between *T*_2_ and the average sputum mesh size (ξ) [158]. Our estimation of CF sputum mesh size was 93 ± 38 nm, in agreement with the findings of Hanes et al. [160]. These authors used micro inert particles (MIPs) of several dimensions to measure the transport rates of MIPs combined with an obstruction-scaling model. Through this approach, they determined that the average 3D mesh spacing of CF sputum was 140 ± 50 nm (range: 60–300 nm). Always based on the diffusion of muco-inert nanoparticle probes in CF sputum, Duncan et al. [161] found that a reduction in sputum mesh size is characteristic of CF patients with reduced lung function. Moreover, to determine the effects of the biochemical components of CF sputum on its microstructure, they quantitatively measured the percent of solid contents (mucin, DNA, and cysteine (disulfide bridge)). The authors observed that the amount of each element negatively correlated with the median MIP transport rate [161].

### 2.2. Mucus in COPD

While CF is a relatively rare disease, COPD is the third cause of death in the Occident [162]. Of the two classical COPD phenotypes, “the pink puffer” (emphysema prevalent) and the “blue bloater” (chronic bronchitis prevalent) [163], the latter exhibits pathologic features similar to CF, including mucin hyperexpression and mucus accumulation. Notably, the chronic bronchitis phenotype affects nearly two thirds of COPD patients [164]. It is now known that exposure to cigarette smoke inhibits CFTR, leading to delayed mucociliary transport and increased mucus viscosity and stasis [164]. Chisholm and coworkers [165] hypothesized that a tightened mesh structure of spontaneously expectorated mucus would contribute to increased COPD disease severity. The authors investigated the mesh size of COPD sputum, quantified through MIP diffusion, in relation to sputum composition and FEV_1_. As a result of their investigation, an inverse correlation was observed between the percentage of solids and MIP, suggesting that a greater solid concentration (for sputum dehydration or mucin hypersecretion) contributes to a tighter sputum mesh in severe COPD. Notably, the mesh size reduction in COPD sputum may also provide a permissive environment for chronic infection and inflammation, slowing down bacteria and neutrophil migration [165] comparably to what observed in CF. Neutrophils are strongly implicated in mucus obstruction due to their release of neutrophil extracellular traps (NETs) upon stimulation. NETs consist of extracellular chromatin networks studded with cytotoxic proteins. When released in the airways, these NETs can become part of the airway mucus [104]. Linssen et al. [166] co-incubated human mucus (collected from endotracheal tubes of healthy subjects) with NETs from phorbol 12-myristate 13-acetate-stimulated neutrophils. Compared to controls, the co-incubation of mucus with NETs resulted in (1) a significant increase in the mucus viscoelasticity (rheology), and (2) significantly decreased mesh pore size of the mucus and decreased movement of MIPs through mucus (micro-rheology). However, NETs did not cause visible changes in the microstructure of the mucus according to scanning electron microscopy. In addition, in the already cited work by Batson et al. [154], neutrophil activity, expressed in terms of elastase levels, presented a correlation with total mucin concentration. Active NE is a serine proteinase secreted by neutrophils in response to inflammation and pathogen invasion. In the literature, NE has been associated with exacerbations, lung function decline, and disease severity [167]. These relationships reflect neutrophil’s activities, which lead to mucin’s increased expression in the airways of COPD patients.

### 2.3. Mucus in Asthma

Asthma is an episodic airways disease characterized by the recruitment of various inflammatory cells to the airways [153]. The persistence of this inflammatory process results in the production of an abnormal mucus that shares properties with CF mucus, including increased mucin levels and high protein concentrations. In its most severe and chronic form, asthma presents increased infectious burden, exacerbation frequency, and poor treatment response, reminiscent of a mild form of CF [168]. In particular, non-allergic asthma has recently been recognized as a CFTR-related disorder [169].

Innes and coworkers [170] analyzed airway mucus from patients in acute asthma exacerbation and from non-asthmatic control subjects through rheological measurements of viscous and elastic moduli and elucidating the microstructure of the fluids, including the degree of cross-linking between protein polymers. Their findings showed that the elastic and viscous moduli were significantly higher than normal. Furthermore, the predominant abnormality in acute asthma was increased cross-linking of mucin polymers (reflected by the markedly increased elastic response), rather than high concentrations of mucins (reflected by the less markedly increased viscous response) [170]. Morgan et al. [171] examined the biophysical properties of airway mucus from asthma patients using the multiple tracking particle (MPT) technique to study whether mucus gel structure, mucociliary clearance, and airway obstruction in asthma can be improved through disulfides disruption of mucus with the reducing agent tris(2-carboxyethyl)phosphine (TCEP). TCEP treatment increased mean-squared displacement (MSD) values, decreased the viscoelasticity of mucus, caused rapid depolarization of mucins, and normalized the homogeneity and level of MSD such that they were similar to those of controls. Another study assessed the contribution of eosinophil and mucin content to mucus rheology from spontaneous unselected sputum samples harvested from patients with asthma and COPD [172]. The asthma group was characterized by elastic and viscous moduli significantly higher (*G*′ = 14.49 Pa·s, *G*″ = 3 Pa·s) than those of the COPD group (*G*′ = 5.01 Pa·s; *G*″ = 1.45 Pa·s). Sputum eosinophil percentages were higher in the asthma group (1.02%) compared to the COPD (0.25%) group, and Spearman’s correlation tests confirmed that *G*’ and *G*’’ were positively correlated with MUC5AC protein concentration [172].

## 3. Mucus Viscoelastic Properties in CRDs

The analysis of the pathological aspects linked to CRDs highlights the fundamental role played by the viscoelasticity of the mucus and its osmotic pressure. These properties have been widely characterized in the literature [173], with reference to healthy and CRD patients [26]. Moreover, the effect of ionic strength and pH has been considered due to their therapeutic implications.

### 3.1. Mucus Viscoelasticity

Viscoelasticity supervises two important biological functions, namely (i) promoting the penetration of nutrients while preventing the spread of foreign particles and pathogens [11], and (ii) assisting the mucus replacement through a proper mechanical response to the stresses induced in mucociliary clearance and coughing [174]. Depending on the scale on which the observation takes place, two different viscoelastic properties can be distinguished: the macroscopic and the microscopic viscoelasticity. Although the latter is more appropriate for describing the local behavior of mucus, both during diffusion and mucociliary clearance, the former is easy to access and can be linked to active particle transport and mucus clearance by coughing [11].

Like all known physical or chemical gels, mucus presents heterogeneity on multiple length scales, possibly as a consequence of its genesis. In fact, the mucins, initially secreted in granular form by the epithelium, undergo a transition into the form of a swollen gel upon exocytosis, following the exchange of monovalent ions with bivalent ones (Na^+^/Ca^2+^) [175]. The gel thus formed behaves similarly to a physical gel from an arrested phase separation, presenting submicrometric heterogeneities between 40 and 500 nm [11,176].

The porosity range and its variation, as a consequence of a particular disease, have a critical role in the diffusivity of molecules and nanoparticles. Depending on the relative length scale and penetrant–mucus interactions, different scenarios can be encountered during mucus penetration [177,178]. When penetrants are smaller than the pore size of the mucus network, the resistance to their diffusion largely reflects the viscous resistance of water, assuming that no specific interactions occur with mucus constituents [179]. However, when the size of the penetrants is increased up to dimensions of about 200 nm, even in the absence of specific interactions, their diffusion rate is reduced compared to water due to the increase in steric hindrance [180]. In other words, the elasticity of the mucus influences the diffusion within it. In ex vivo or in vitro experiments, the resistance offered by the elasticity of the mucus to particles in the above range is such that free diffusion is practically inhibited. However, in the case of in vivo measurements, periodic mechanical loads applied to the mucus layer by the cilia can lead to a local dynamic change in pore size and viscoelastic properties [177], leading to local fluidization [181] and making penetration feasible. Such a situation is mostly expected when the particle diameter is similar to the pore size. Finally, on length scales significantly greater than 500 nm, the steric hindrance is such that it prevents diffusion, regardless of the experimental conditions adopted (i.e., in vitro or in vivo) and/or the interaction between particles and mucus. Figure 4 schematically summarizes the abovementioned scenarios.

The above scenario changes for CRD patients show an increase in elastic modulus (*G*′) and viscous or loss modulus (*G*″) and a reduction in mesh size [182]. Studies conducted by Hanes and co-workers with non-muco-adhesive nanoprobes suggest that the micro-viscosity of fresh, undiluted sputum from CF patients is greater than that of water by a factor of only about 5 [183], while, conversely, the variations in macroscopic viscosity compared to the sputum of healthy subjects are 10,000 times greater. Plausibly, sputum from CF patients shows greater heterogeneity of the mucus network, resulting in larger pores for particle diffusion and greater macroscopic viscosity and elasticity. This ambivalence is clear evidence of the hierarchical structure of mucus, which involves a variation between macroscopic and microscopic viscoelasticity. A finding in line with the idea of elasticity-assisted diffusion in in vivo samples shows that there is a positive correlation between variations in the elastic modulus (*G*′) of CF sputum samples and particle diffusion of approximately 100–200 nm (*G* > 100 Pa·s) [184].

As mentioned in Section 2.1, in addition to presenting a hyperconcentration of macromolecular material, CRD patients also present an increase in DNA and other proteins. The presence of non-mucin macromolecules has been shown to have a strong influence on the cohesion of the mucus network [185,186]. In the case of chronic lung diseases, such as CF, COPD, and asthma, an increase in the macroscopic viscoelasticity of the mucus is observed due to reduced water content and an increased fraction of high-molecular-weight material other than mucins (such as DNA and proteins) [187,188]. For example, CF sputum is significantly more elastic than viscous at all frequencies, with a value of loss tangent (equal to the ratio between *G*″ and *G*’) $tan\left(\delta \right)=0.29÷0.31$ at a cutoff frequency of 1 rad/s. Correlations between macro-rheological properties and disease have been demonstrated both in the case of CF [60] and for neonatal respiratory distress syndrome [189].

Some variability in macro- and micro-viscoelastic properties is, however, physiological, because it allows for the adaptation of mucus clearance in response to environmental stimuli [190]. For example, when irritants are encountered, such as cigarette smoke or particular allergens, more dilute mucus is secreted, thus decreasing viscoelasticity for faster clearance [191,192]. Nevertheless, in cases of prolonged exposure to irritants or in cases of chronical diseases, mucociliary clearance is subjected to a radical change [193].

Mucociliary clearance is a crucial step in maintaining an effective defense of the airways against viral and bacterial infections. Layers of mucus are, in fact, constantly removed by the mobile cilia (peristaltic movement) and through mechanical forces (coughing), resulting in a continuous flow of mucus on the airways. The average clearing times are approximately 20 min for the airways of a healthy person [194]. An appropriate range of viscoelastic properties, both macro and micro, is also essential for mucociliary clearance [195,196]. The mucociliary clearance simulations conducted by Chatelin et al. showed the effect on the clearance rate of the changes in nonlinear viscoelastic properties. In particular, the shear-thinning rheological property of the mucus, described in the Chatelin model through its rheological indices (i.e., flow consistency index k and power law index n), has an effect on both mucociliary clearance and coughing [197].

When mucociliary clearance is impaired, coughing becomes the primary mechanism of mucus replacement. In cough clearance, the nonlinear viscoelasticity of mucus and, in particular, its yield stress plays a fundamental role. This quantity is accessible through nonlinear measurements, such as the large-amplitude oscillatory shear (LAOS). In general, for healthy subjects, the yield stress has values ranging from 0.1 to 1 Pa·s, while it can reach 5–50 Pa·s for subjects suffering from serious pathologies. A calculation based on the air coming out during coughing leads to an estimate of the force acting on the mucus as 100 Pa·s. This value places an upper limit on the yield stress that a thick mucus can reach while still allowing for expectoration of the mucus.

Collectively, these data show that CRD pathologies present a common hyperconcentration of mucus and an alteration of its composition, which makes the mucus more viscous and more elastic. Such characteristics limit the diffusion of therapeutic agents and prevent mucus clearance. Micro- and macro-rheology, both in linear and nonlinear regimes, are still useful tools for revealing the relationship between viscoelastic properties and the composition and structure of mucus, a topic largely unexplored.

### 3.2. Osmotic Pressure

In addition to the viscoelastic properties, a fundamental role in the physiology of mucus is played by the osmotic pressure of the mucus and the periciliary fluid layer (PCL). The gel-on-brush model developed by Rubinstein and co-authors describes the mucus plus PCL system as a brush-like structure occupying the gaps between the cilia and topped by a layer of mucus gel [145] (Figure 5B). The presence of a brush-like periciliary layer prevents the penetration of mucus between the cilia and allows the mucus to form an independent layer. The relationship between the osmotic modules in the two layers explains the different behaviors of mucociliary clearance for healthy and CRD subjects.

The cartoon in Figure 5 depicts the three conditions in which the supra-epithelial layer can be found in different kind of patients, and it illustrates the effects of the osmotic pressure of the mucus gel and the *PCL* on mucociliary clearance. Under physiological conditions, the osmotic modulus of mucus, represented by a green spring (K_mucus_), is lower than that of the *PCL* (K_PCL_) (Figure 5B). In this case, water added to the healthy airway surface dilutes the mucus layer, leaving the *PCL* unchanged (Figure 5A). Under a dehydrated state, instead, when water is removed from the airway surface, it preferentially leaves the mucus gel first, due to its lower osmotic modulus. However, further dehydration (Figure 5C) leads to the removal of water from both the mucus gel and the PCL. The modules of both layers are increased and equal, represented by smaller diameters of shortened springs. This state corresponds to diseased airways (COPD and CF).

The mucus of CRD patients shares the same characteristics, including hyperconcentration. As demonstrated by Boucher and co-authors [198], the sputum obtained from these patients presents both a higher percentage of solids and a higher concentration of total mucins. The increase in the solid fraction leads to a corresponding increase in partial osmotic pressures. Because mucociliary clearance is closely linked to the partial osmotic pressure of the mucus [145], it follows that in these patients a negative correlation exists between the mucociliary clearance and the solid fraction present in the mucus.

In summary, the osmotic pressure of the mucosa is a fundamental parameter for correctly interpreting the pathologies of patients suffering from CRD, such as mucus stasis. An effective medical therapy, in addition to reducing the increased viscoelasticity of the mucus, must therefore necessarily aim to rehydrate and restore the osmotic properties of the mucosa.

### 3.3. The Role of pH and Ionic Strength in Mucus Properties

Among the different conditions that influence the properties of mucus gel, particular attention was paid to the effect of ionic strength and pH [199,200,201]. Indeed, the variety of interactions between the constituents of mucus (mucins and non-mucin proteins) causes the variation in the concentrations of cations and protons to have an effect on its viscoelastic properties. Hydrogen bonds, hydrophobic and ionic interactions, and covalent bonds (disulfide bonds) have all been mentioned as responsible for the formation of the mucin network and therefore sensible to change in the ion composition [135,202]. From a chemical–physical point of view, these effects can be ascribed to the polyelectrolytic nature of the mucus components [203,204]. In fact, an increase in pH, as well as an increase in ionic strength, causes a destabilization of the electrostatic interactions, which leads to variation of the macroscopic viscoelastic properties.

Regarding proton concentration, it was ascertained that a reduction in pH causes an increase in the viscoelasticity of ex vivo mucus, both in healthy and CRD subjects [200,205]. However, more recent evidence has downplayed the effect such variations have on patients with chronic diseases. For instance, changes in pH relevant to CF induce a relatively weak change in the biophysical properties of in vivo mucus compared to those induced by an increase in the solid fraction [59].

Similarly, the addition of monovalent cations (Na^+^) has an effect both on the electrostatic interactions between the macromolecular components of mucus and on the competition between monovalent and divalent ions. In the latter case, in fact, Na^+^ replaces the Ca^2+^ ions present in the initial mucus granule, thus interrupting the salt bridges between mucins. Last but not least, the Na^+^ ion acts as an osmotically active solute, leading to the rehydration of hyperconcentrated mucus [206].

Changes in ASL pH have been particularly investigated in CF. ASL in CF piglets is acidic due to reduced CFTR-dependent bicarbonate transport at the level of the surface epithelium [97,207], likely causing higher susceptibility to *S. aureus* colonization and infection [207]. Interestingly, a *Cftr*^−/−^ rat model developed hyperacidic ASL, and dehydration of the airway surface imparted mucus stasis unless gland secretions were mature. Moreover, aberrant mucus transport could be corrected by adding bicarbonate to normalize mucus viscosity, indicating that mature airway secretions are required to manifest the CF defect primed by airway dehydration and bicarbonate deficiency [208]. Overall, these data may help in understanding the complex interplay of PCL depletion, ASL hyperacidic pH, and mucus secretion by submucosal glands at the onset and throughout the progression of lung disease in humans.

Knowledge of the effects of ionic strength and pH has several therapeutic implications. The former is linked to treatments with hypertonic solutions so as to restore the hydration of the ASL [209]. The latter, instead, is important in the antimicrobial defense of patients suffering from CF, as described elsewhere in this review. However, the above reported effect of bicarbonate on mucus viscosity indicates how further research on this aspect of mucus pathophysiology is warranted.

## 4. Mucus/Sputum Mediators as Biomarkers in CRDs

Mucociliary clearance and cough represent the major defense mechanisms in the conducting airways. On the other hand, mucus is replenished by antibacterial substances, such as enzymes (e.g., lysozyme, secreted phospholipases), polypeptides (lactoferrin, secretory leukoprotease inhibitor (SLPI)), and peptides [210,211,212,213]. Antimicrobial peptides (AMPs, chiefly defensins and cathelecidins) are a first line of innate immunity, being secreted by airway epithelial cells and neutrophils in an early phase of infection and with rapid kinetics. Mucus constituents may modulate the activity of AMPs. Indeed, they can bound to mucins [214], bacterial polysaccharides released by lung pathogens [215], and DNA and F-actin [216]. Variable levels of these mediators have been found in the airway secretions, including sputum, due to their interactions with other constituents or the clinical status of CRD patients. CF sputum contains high levels of the only known human cathelecidin, LL-37 (and its precursor, hCAP-18), and mucins at higher concentrations than BALF [214], while another study found higher hCAP-18 levels in sputum from CF patients compared with healthy control subjects [217]. LL-37 was found in CF sputum bound to various components, such as DNA, F-actin, and cellular debris, as well as to LPS, deploying for its decreased antimicrobial activity [218]. LL-37 was found to be complexed to glycosaminoglycans in CF airway secretions (BALF and sputum), but it is liberated following nebulized hypertonic saline, resulting in an increased antimicrobial effect [219]. Lysozyme, lactoferrin, SLPI, and hCAP-18/LL-37 are elevated in sputum from stable COPD patients compared with healthy control patients [217,220,221,222,223]. While sputum hCAP-18/LL37 increases during COPD exacerbation [224], levels of the antiprotease SLPI decrease during acute exacerbations in patients with COPD and rise after antibiotic therapy [225,226]. A complex picture emerges from the study of different sputum antimicrobial substances in COPD exacerbations associated with nontypeable *Haemophilus influenzae* (NTHI) and *Moraxella catarrhalis*; compared to the baseline, lysozyme and SLPI lowered, LL-37 increased, and lactoferrin remained unchanged [227]. COPD acute exacerbations induced by rhinovirus infection were characterized by higher bacterial burden and low levels of sputum levels of the AMPs SLPI and elafin, suggesting that either their decrease increases susceptibility to bacterial infections or sputum NE is responsible for their degradation [228]. In persistent asthma, sputum hCAP-18 levels were significantly reduced compared to control subjects or patients with CF or COPD [217]. Sputum *β*-defensin-1 (hBD-1) was significantly higher in severe asthma compared to controlled and uncontrolled asthma; however, hBD-1 was significantly higher in COPD subjects than in asthma subjects and healthy controls [229].

Sputum biomarkers have been investigated, either alone or in combination with other matrices, for diagnostic, prognostic, and therapeutic purposes, and thus we refer the reader to extensive review articles (CF [230,231,232,233], COPD [234,235,236,237,238], asthma [234,236,239,240]); for this reason, the literature analysis focused on the last five years (2019–2023).

### 4.1. Cystic Fibrosis (Table 2)

The percentage of predicted forced expiratory volume in 1s (ppFEV_1_) is an established measure of disease severity in CF that is used to capture therapeutic efficacy [241,242], and thus it is considered the benchmark for biomarker validation [243]. In comparison to BAL, bound to invasive maneuvers and disease exacerbation [244,245], sputum is a valid option to explore local airway biomarkers, especially after induction, which carries minimal risk, and it is possible in pediatric populations [246] and those unable to expectorate [247,248]. Among inflammatory mediators in CF sputum, neutrophil elastase (NE) has received the most attention and was identified as the most sensitive biomarker of lung disease severity and as a predictor of the clinical outcome [243,249,250,251]. By comparing an ELISA assay and two enzymatic digestion assays (chromogenic –CS– and fluorogenic –FS– substrate), Oriano and colleagues [252] found that NE activity in spontaneous sputum measured through FS and ELISA performed similarly in predicting chronic *P. aeruginosa* infection (AUC of 0.73 and 0.75, respectively) and that the highest correlation between active NE and ppFEV_1_ was found with FS. These data suggest the appropriateness of these two techniques compared to the CS method for CF. However, no correlations of NE levels using any method correlated with sputum mucopurulence.

A study went across three retrospective CF cohorts spanning a wide range of diseases to explore the correlation of serum and sputum resistin with lung disease severity [253]. Resistin, a ligand for Toll-like receptor 4, modulates the recruitment and activation of neutrophils via the NF-κB signaling pathway [254]. A combined analysis across all patient cohorts revealed a strong negative correlation between sputum resistin and lung function [253]. On the other hand, Reihill and colleagues [255] focused on a single marker, i.e., serine trypsin-like proteases (TLP), which are involved in ENaC activation and mucus overproduction in CF airways, potentially through the activation of protease-activated receptor-2 [256,257]. In one cohort, TLP activity inversely correlated with lung function (ppFEV_1_), TLP higher activity was evident in individuals who did not survive beyond 5 years, and individuals with high (above median) TLP activity had a poorer survival outcome compared with individuals with low (below median) TLP activity. In the other cohort, it was just possible to show a nonsignificant trend [255], likely for the different clinical status of the two groups (one stable and the other hospitalized for exacerbation).

A multicenter study of a CF cohort homozygous or heterozygous for the *F508del* mutation spanning children, adolescents, and adults investigated whether induced sputum biomarkers were more descriptive of the pulmonary disease than serum biomarkers [258]. At baseline, biomarkers that correlated most strongly with ppFEV_1_ were serum high sensitivity C reactive protein (hsCRP), serum amyloid A (SAA), and sputum NE. At the 16-week time point of observation, the changes in biomarkers that correlated most strongly with a change in ppFEV_1_ were serum hsCRP and sputum NE. Serum hsCRP and MPO and sputum NE, TNF-α, IL-8, and MPO were significantly higher in adults (≥18 years) compared with children and adolescents (<18 years). Moreover, all of the serum and sputum biomarkers were significantly higher in those with lower ppFEV_1_ (<80%) compared with those with more preserved ppFEV_1_ (>80%). Interestingly, there were no significant differences in paired measurements of NE, IL-1β, and MPO in the sputum aliquots that underwent immediate processing (“fresh”) compared to the sputum aliquots that were processed after freezing. We have previously shown that there were no significant effects of freezing and thawing on measurements of IL-8 and TNF-α in sputum supernatant that had already undergone partial homogenization and centrifugation to remove host cells [259].

Through an exhaustive literature search and a systematic review of inflammatory biomarkers in induced sputum across 139 included articles, Lepissier et al. identified 71 sputum biomarkers to undertake evaluation of their clinimetric properties (across potential pharmacodynamic, monitoring, predictive, and prognostic roles), responsiveness, discriminant (against healthy controls and other respiratory diseases), concurrent, and convergent (against ppFEV_1_) validity [232]. Among the biomarkers studied, only IL-8 showed significantly higher levels in CF versus asthmatic patients, whereas variable discrimination was found between CF patients and patients with bronchiectasis and COPD. Concurrent validity was inconsistent for a range of clinical outcomes, including ppPFEV_1_ decline, survival, bacterial or pulmonary exacerbation status, symptoms, and radiological appearances. Overall, however, NE, IL-8, TNF-α, and IL-1β demonstrated validity and responsiveness to therapies to be currently considered for use in clinical trials. Other biomarkers, such as IL-6, calprotectin, HMGB-1, and chitinase 3 like protein YKL-40, demonstrated potential utility.

Liou et al. [260] asked whether sputum inflammatory biomarkers provide explanatory information on pulmonary exacerbations. Of the 24 inflammatory markers investigated, 10 were plausibly associated with time to next exacerbation (see Table 2). Three potential biomarkers of RAGE axis inflammation (ENRAGE, sRAGE, and calprotectin) were associated with time to next exacerbation, while six potential neutrophil-associated biomarkers indicated associations between protease activity or reactive oxygen species (MPO, ICAM1, NE, YKL40, TARC, MMP9) and time to next exacerbation. However, only MPO and ENRAGE could be considered the most solid biomarkers in statistical analysis (False Discovery Rate > 99%). Following treatment with ivacaftor or ivacaftor–lumacaftor, individual biomarker variable effects were stable throughout testing.

Although individual proteins relate to the severity of lung diseases, as determined through baseline ppFEV_1_ [243,249,261,262,263,264,265,266,267], a proteomic analysis, i.e., the study of large-scale changes in protein expression, can reveal molecular pathways of disease and provide translational perspectives to inform clinical decision making. Proteomic analysis of CF sputum samples has established that the CF respiratory tract proteome is distinct from healthy controls, with the differences largely driven by upregulation of neutrophil-related immune and inflammatory responses, such as MPO, IL-8, and calprotectin (calgranulin A and B), as well as differences in a number of pathways, including those related to oxidative stress and actin cytoskeleton rearrangement [261,264,268]. By using a sputum proteomic approach, Maher et al. [269] recently found that 20 proteins discriminated most between CF and healthy control samples. At baseline, 87 proteins correlated with ppFEV_1_, 67 of which increased with worsening lung function (discriminated by ppFEV_1_ stratification), while 20 showed the inverse relationship. In general, the molecular functions and biological processes associated with all of these 87 proteins were in keeping with progressively increasing neutrophil activity and greater disturbance of imbalances in protease/anti-protease and oxidant/antioxidant states. Among the proteins that increased, the authors found (1) those having previously been shown to relate to CF lung disease severity or increase during pulmonary exacerbations, such as myeloperoxidase, azurocidin, chitotriosidase-1, protein S100-A8/A9, cathepsin G, and matrix metallopeptidase 9; (2) those involved in neutrophil extracellular traps (NETs), e.g., NE, histones H2A and H2B, catalase, and glucose-6-phosphate isomerase, and (3) those neutrophil-derived and implicated in the innate immune response (triose phosphate isomerase, peptidoglycan recognition protein 1, and phospholipase-B-like-1). Among those that decreased with increased lung severity, anti-oxidants (lactoperoxidase, glutaredoxin-1), anti-inflammatory and immunomodulatory (alpha-1-antichymotrypsin, interleukin-1 receptor antagonist protein) were the most notable. Interestingly, all CF subjects taking ivacaftor overlapped with healthy controls irrespective of underlying lung function impairment.

Nowadays, as CFTR modulator therapy is a current clinical practice, some data are also emerging on the modulation of sputum biomarkers and its properties. Previous studies highlighted a reduction in sputum inflammatory markers [270] and a reversal of altered viscoelasticity [271]. CFTR modulator therapy decreased mucin concentration, produced a relaxation of the mucin network, and increased mucin removal via hydration, and not pH or HCO_3_^−^, in primary airway epithelial cells with either *G551D* or *F508del* mutation [272]. In the study by Harris et al., 6-month ivacaftor (IVA) treatment of CF patients with at least one *G551D* mutation did not change sputum inflammatory marker levels in the overall study cohort [273]. This lack of difference before and after therapy was maintained when two subgroups were analyzed, i.e., one with milder lung disease assessed through sputum induction and a second with more impaired lung function who spontaneously expectorated sputum.

Greaber et al. [274] aimed at determining the effects of lumacaftor (LUMA)–IVA on novel endpoints of lung ventilation (Lung Clearance Index, LCI), lung structure, and perfusion (Magnetic Resonance Imaging, MRI), as well as airway infection and inflammation in patients with CF homozygous for the *F508del* mutation in a post-approval setting. LUMA-IVA therapy significantly reduced sweat chloride concentration and increased body mass index (BMI), whereas ppFEV_1_ increased, but not significantly. Ventilation (LCI) was improved as morphological/perfusion abnormalities were detected using MRI. While sputum IL-1β concentration was significantly reduced to about 25% of baseline values after therapy, IL-6, IL-8, TNF-α, and free NE activity remained unchanged after starting LUMA–IVA therapy. Of note, besides acting as a neutrophil chemoattractant, IL-1β has been revealed as a potent stimulus for mucus hypersecretion by the airway epithelium [115], with its decrease due to LUMA-IVA likely contributing to inducing mucus plugging of the airways.

In a real-world post-approval study [275], CF patients were evaluated for plasma and sputum inflammatory biomarkers before and after 12 months of LUMA–IVA treatment. In conjunction with an increase in ppFEV_1_ and BMI, as well as a reduction in sweat chloride concentration, intravenous antibiotic courses, and the mean number of exacerbations per patients, significant reductions in sputum IL-6, IL-8, IL-1β, and TNF-α before and after 3 and 12 months of treatment were observed. Significant and sustained reductions in plasma IL-8, IL-1β, and TNF-α were also revealed at 1 year of treatment.

Schaupp et al. [276] investigated in a longitudinal study the effects of ETI (elexacafor/tezacaftor/IVA) on sputum rheology and key inflammatory markers at baseline and at 1, 3, and 12 months after the initiation of ETI therapy. ETI improved sputum viscoelastic properties, as judged by the reduction of the elastic modulus (*G*′) and viscous modulus (*G*″) as well as by the increase in the mesh pore size (ξ). These changes became significant at 3 months for *G*′ and *G*″ and at 12 months for the mesh pore size compared to the baseline. However, *G*′ and *G*″ remained significantly elevated, and the mesh pore size was still decreased after 12 months of ETI therapy in comparison with healthy controls. ETI therapy determined a lowered level of sputum IL-1β at all time-points compared to baseline. Although IL-8 showed a reduction at 3 and 12 months, these changes were not significant. IL-1β, IL-8, IL-6, and TNF-α were still elevated after ETI in comparison with healthy controls. ETI significantly reduced free NE activity at 1, 3, and 12 months. Measurement of protease/antiprotease balance through proteomic analysis revealed that cathepsin G and proteinase 3 (PR3) were reduced while secretory leukocyte protease inhibitor (SLPI) was increased and α_1_-antitrypsin (AAT) was reduced following ETI therapy. Free NE activity, as well as cathepsin G, PR3, and AAT levels, were still elevated in CF sputum samples at 12 months of therapy in comparison with healthy controls. Finally, ETI therapy shifted sputum proteome toward that of healthy controls.

Metabolites identified in various matrices (exhaled breath condensate (EBC), sputum, saliva, plasma, and urine) may have clinical potential for predicting or diagnosing pulmonary exacerbation (PEX) in CF [277]. Pyruvate, lactate, and putrescine showed significant increases in expectorated sputum samples from patients in exacerbation compared to stable patients, whereas palmitate levels decreased [278]. On the other hand, Hanusch et al. [279] have recently demonstrated that sputum arginine was higher in PEX (due to a bacterial infection) compared to stable CF samples and that arginine and homoarginine dropped after antibiotic treatment. These data were interpreted as either a compensatory mechanism in elevating nitric oxide during acute inflammation in infected patients or increased bacterial arginine synthesis and secretion; in both cases, antibiotics were effective in normalizing arginine lung levels to pre-exacerbation levels.

Classification of CF patients in two groups as sustained symptom-responders (SRs) or non-sustained symptom-responders (NSRs) to treatment for a PEX allowed for the study of serum and sputum iron as a biomarker for this response difference [280]. SRs and NSRs had no significant differences in lung function at pulmonary exacerbation onset and treatment durations. SRs showed serum iron increased over time and sputum iron was relatively stable, whereas NSRs displayed serum iron that initially increased but then decreased, while sputum iron first decreased and then increased around day 7 of treatment. It has to be determined whether this increased sputum iron level associated with unfavorable clinical outcomes of PEX treatment is correlated somehow with *P. aeruginosa* infection, as suggested by some studies [281].

Levels of hBD-2 and hCAP-18/LL37 in CF sputum have been demonstrated to be higher than in control specimens [282], confirming previous results regarding the lack of correlation of increased AMPs levels and *P. aeruginosa* infection when AMPs were studied in BALF [283]. Together with the results obtained by Xiao et al. [217], overall, these data indicate that sputum is an easily accessible and representative sample of airway secretions for the measurement of AMP levels.

**Table 2 ijms-25-01933-t002:** Mucus/sputum biomarkers in CF.

Patient Cohorts	Sputum Samples	Outcomes	Reference
Three retrospective CF cohorts spanning a wide range of diseases	Induced or spontaneous sputum	Sputum resistin levels were negatively correlated with CF lung function	Forrest et al., 2019 [253]
CF patients (n = 47) with ppFEV_1_ of 67.5 (19.3) (mean (SD))	Spontaneous sputum	NE activity correlates with chronic *P. aeruginosa* infection according to ELISA and fluorogenic assays, while NE activity correlates with ppFEV_1_ according to the fluorogenic assay	Oriano et al., 2019 [252]
CF patients (n = 31) 10 years and older, with at least one *G551D* CFTR allele	Induced or spontaneous sputum	IVA treatment did not result in significant changes in inflammatory marker levels (Il-1β, IL-6, IL-8, NE, SLPI, and AAT)	Harris et al., 2020 [273]
CF patients (n = 70), median age 11.8 years	Induced sputum	During acute exacerbation, sputum arginin and homoarginine levels were high and dropped after antibiotic treatment	Hanusch et al., 2020 [279]
Adult CF patients in two study cohorts	Spontaneous sputum	Serine trypsin-like proteases (TLP) activity inversely correlated with ppFEV_1_. Individuals with high TLP activity showed significantly reduced survival	Reihill et al., 2020 [255]
CF patients (n = 30) 12 years and older homozygous for the F508del mutation	Spontaneous sputum	LUMA-IVA decreased IL-1β levels after 8–16 weeks of treatment	Graeber et al., 2021 [274]
CF patients (n = 73) 10 years of age or older and with ppFEV_1_ of 74.1 (14.4) (mean (SD))	Induced sputum	At baseline and after 16 weeks, NE negatively correlated with ppFEV_1_	Jain et al., 2021 [258]
CF patients (n = 20) 29.9 (8.5) years with ppFEV_1_ of 49 (22) (mean (SD))	Spontaneous sputum	For sustained-symptom responders, sputum iron was relatively stable, while for non-sustained symptom-responders, sputum iron first decreased and then increased around day 7 of treatment	Gifford et al., 2021 [280]
CF patients across 139 articles and 71 biomarkers	Spontaneous and induced sputum	NE, IL-8, TNF-α, and IL-1β demonstrated validity between CF and non-CF subjects, as well as responsiveness to therapies	Lepissier et al., 2022 [232]
Randomly chosen clinical stable CF patients (n = 114) 28 (12) years with ppFEV_1_ of 70 (22) (mean (SD))	Spontaneous sputum	Biomarkers associated with time to next exacerbation: ENRAGE, MPO, sRAGE, ICAM-1, NE, YKL40, TARC, MMP-9, IL-1β, IL-5	Liou et al., 2022 [260]
CF patients (n = 38) >18 years of age	Spontaneous sputum	Proteomic analysis revealed that baseline ppFEV_1_ severity correlated with 87 proteins; positive correlation n = 20, negative n = 67); most were either neutrophil-derived or involved in neutrophil-driven oxidant and protease activity	Maher et al., 2022 [269]
CF children (n = 27) with median age of 11.4 years	Spontaneous sputum	There were higher levels of hBD-2 and hCAP-18/LL-37 in the study group compared to the control group	Ishchenko et al., 2022 [282]
CF patients (n = 44) aged 16 years and older homozygous for the F508del mutation	Spontaneous sputum	After 12 months of LUMA–IVA treatment, significant reductions in sputum IL-6, IL-8, IL-1β, and TNF-α levels were observed	Arooj et al., 2023 [275]
CF patients (n = 79) aged ≥12 years with one or two F508del alleles	Spontaneous sputum	ETI improved the elastic modulus and viscous modulus of CF sputum at 3 and 12 months after initiation, reduced IL-8 and free NE activity, and shifted the CF sputum proteome toward healthy	Schaupp et al., 2023 [276]

ENRAGE: extracellular newly identified receptor for advanced glycation end-products; ETI: elexacaftor/tezacaftor/ivacaftor; ICAM-1: intercellular adhesion molecule-1; IL: interleukin; IVA: ivacaftor; LUMA: lumacaftor; MMP-9: matrix metalloprotease-9; NE: neutrophil elastase; ppFEV_1_: percent predicted FEV_1_; sRAGE: soluble receptor for advanced glycation end-products; TARC: thymus and activation regulated chemokine; TLP: serine trypsin-like proteases; YKL40: chitinase-3-like 1 protein.

### 4.2. COPD (Table 3)

The diagnosis and follow-up of COPD are currently conducted using spirometry as the gold-standard method [284]. However, this methodology is highly dependent on specialized operators for evaluation of the results, the quality of the device, which can be incorrectly calibrated or contain leaks, and the condition of the patient (spirometry is inadvisable for patients with cardiac problems or dementia) [285]. Lack of outcome prediction is also reported to be a vital disadvantage of spirometry; therefore, it is inefficient to prevent acute exacerbations. Thus, the possibility of monitoring concentrations of biomarkers in bio-fluids in association with clinical parameters (weight, age, and acute exacerbation frequency) would improve the follow-up and outcome prediction of patients diagnosed with COPD [286]. On the basis of their application, it is possible to classify biomarkers as screening, diagnostic, or prognostic markers depending on their ability to predict, diagnose, or assess disease status, respectively [287,288]. In general, a typical COPD patient with a decades-long smoking history will show the classical symptoms in the last years of their life, and this dictates the need to identify new prognostic biomarkers, such as those superior to spirometry, in evaluating prognosis. Promising prognostic biomarkers should have some characteristics to be applied in clinical practice, such as the possibility to discriminate among the different clinical phenotypes and the ease of collection.

In these last years, the use of proteomics platforms has made many contributions in the field of biomarkers discovery, allowing a better interpretation of COPD pathophysiology. Terracciano et al. [289] performed, by means of innovative technology, a comparison of the proteomic/peptidomic profiles of sputum from patients affected by COPD and asthma. This technology optimized the stage of sample preparation through modified mesoporous silica beads with high stability. The beads used in the study were characterized by physical features (pore size distribution, high pore volume, and high surface) that improved the acquisition of specimens, since allowing the acquisition of highly enriched and well-resolved MALDI-TOF peptide profiles. Interestingly, these researchers identified six signals corresponding to three human α-defensins and three C-terminal amidated peptides (one of which was phosphorylated on serine), which could be useful to differentiate these inflammatory airway diseases. In a review published more recently, Dong et al. [237] focused their attention on some protein biomarkers (in saliva and sputum) that could be useful in the management of COPD. More precisely, they evaluated the clinical feasibility of interleukins (IL-6 and IL-8), matrix metalloproteinases (MMP-8 and MMP-9), C-reactive protein, TNF-α, and neutrophil elastase. Furthermore, these authors evidenced that these biomarkers in saliva and sputum could be involved in the development of point-of-care sensors for chronic lung disease management. Dasgupta et al. [290] carried out a pilot study to characterize the proteomic profile of sputum from patients with different lung diseases, such as asthma, COPD, and chronic bronchitis. In that study, the role of eight proteins (Azurocidin1, Neutrophil defensin 3, Lactotransferrin, Calmodulin 3, Coronin1A, Mucin 5B, Mucin 5AC, and BPI fold containing family B1) was investigated. Interestingly, they observed that these proteins could be useful in discriminating between exacerbator/non-exacerbators patients. Furthermore, the study has shown for the first time that the total protein concentration could be a useful biomarker to identify the “frequent exacerbators”. These findings will help in the identification of COPD patients with exacerbations. In that study, the sample size was limited; therefore, larger studies are necessary to validate these results.

Recently, a study by Barta et al. [291] evaluated the expression of 120 cytokines in the sputum of healthy individuals, stable COPD patients, and those experiencing a severe exacerbation using protein microarray. Interestingly, the study investigated if the cytokine profiles of patients with exacerbation can be modified by treatments received during hospitalization. It is interesting to observe that compared to healthy controls, stable COPD patients were characterized by an increased expression of two cytokines: IL-6 and GROα. These results are consistent with the role that these two mediators exert in the pathogenesis of COPD. GROα is a chemokine that plays a key role in the recruitment of inflammatory cells, especially neutrophils, while IL-6 is another inflammatory mediator responsible for many systemic features of the disease [292]. Moreover, several cytokines were altered at the onset of exacerbation and decreased after treatment. In particular, two of these mediators, RANTES and M-CSF, are important regulators of the inflammatory response, as the first one is a chemotactic agent for neutrophils and/or eosinophils [293], while M-CSF regulates both the differentiation and survival of inflammatory cells [292]. It must be pointed out that many cytokines did not return to the levels of stable patients, which may be because the duration of hospitalization (7–14 days) was not enough to allow this recovery, indicating that recovery from exacerbation should be tested for a longer time.

An interesting study devoted to identifying which physiologic pathways are altered in the airways of patients with COPD and if there are further markers of disease severity was published by Esther and colleagues [294]. In this work, mass spectrometry was used to characterize the supernatants of sputa from patients with COPD. Of note, they observed that sialic acid (a molecule correlated to the level of mucus hydration) and two metabolites of adenosine (hypoxanthine and xanthine) could be useful biomarkers to assess disease severity, to predict exacerbations, and to evaluate the efficacy of new drugs. Interestingly, in patients with a greater spirometric severity, the increase in sialic acid from two-fold to three-fold was indicative of a level of mucus hyperconcentration that causes a reduced mucus clearance and, in turn, airflow obstruction, inflammation, and infection. Moreover, the study demonstrated a positive correlation between sialic acid and GOLD status, between mucus concentration and airflow obstruction, and between sialic acid and neutrophil numbers, with the latter evidencing a role for mucus stasis and accumulation in perpetuating airway inflammation. Previous studies demonstrated that many airway inflammatory diseases [295,296] are characterized by an increase in the metabolism of extracellular airway adenosine, which results in the production of hypoxanthine and xanthine. These two metabolites are substrates for xanthine oxidase, which contributes to oxidative stress [297]. It is interesting to observe that in normal conditions, the clearance of noxious agents, such as cigarette smoke, is granted by a coordinate release of mucin and adenosine. This process is due to the co-packaging of mucins with adenosine precursors (adenosine diphosphate and the adenosine metabolic pathway) that are metabolized on airway surfaces to adenosine, as well as nucleotide release via other mechanisms [298,299,300]. On the contrary, in the presence of COPD progression, increased mucin secretion rates were accompanied by increased extracellular adenosine metabolism, preventing sufficient adenosine-mediated mucus hydration. Even if the cause of the increased adenosine metabolism in COPD was not determined in that study, it is reasonable to argue that the epithelial cell reprograming or the action of airway macrophages or infiltrating neutrophils could have a role. Because patients with elevations in both sialic acid and hypoxanthine were at the highest risk for future pulmonary exacerbations, the authors conclude that adding these biomarkers to the clinician’s toolbox could enhance the predictive ability of established models.

In general, the detection of airway inflammation in COPD sputum is carried out using immunoassays. There is a very poor literature concerning their performance and validation in this complex matrix. Recently, Mulvanny et al. [301] published a study about the validation of sputum supernatant biomarkers as well as their practical utility in COPD clinical studies. In the first part of the study, candidate biomarkers were validated through a series of experiments. Then, validated biomarkers were used to assess sputum supernatants in Cohort A (n = 30 COPD, n = 10 smokers, n = 10 healthy) and Cohort B (n = 81 COPD, n = 15 smokers, n = 26 healthy). Cohort A sputum supernatants were analyzed using a 3-plex Luminex multiplex Assay (Merck Millipore, MA, USA), while Cohort B sputum supernatants were analyzed using a 27-plex Luminex multiplex assay (Bio-Rad, Hertfordshire, UK). It is noteworthy to observe that the results for the common cytokines studied (IL-1β, IL-6, TNF-α, and IL-8) were not identical across cohorts, which can be explained by both differences between immunoassays and differences in cohort characteristics. In that study, the comparison of immunoassays was not conducted on a single cohort because the volume of each sputum supernatant sample was not enough to test all immunoassays. Interestingly, COPD sputum neutrophil percentage counts were higher in Cohort A versus B (83.7% and 68.8%, respectively). Some of the 3-plex cytokines (IL-1β and TNF-α) were positively correlated with neutrophil percentages in Cohort B, so similar neutrophil counts between COPD patients and controls in Cohort B reduce the possibility of elucidating between-group differences for these cytokines. It must be said that previous studies have shown conflicting results for IL-1β and TNF-α, with increased levels in COPD patients and no difference versus controls being reported [302,303]. Nevertheless, despite the differences in clinical and immunoassay characteristics between cohorts, there were significant differences between COPD patients and controls for IL-6 and IL-8 in both cohorts, consistent with previous studies [302,303,304,305]. They also showed statistically significant increases in IL-1β, IL-6, and TNF-α during an exacerbation, as previously reported [306,307].

As for antimicrobial proteins and peptides, it has been previously shown that sputum levels of SLPI and elafin display an inverse correlation with bacterial load [228]. In a more recent study in COPD and asthmatic patients [308], an overall negative correlation between both sputum SLPI and hBD-1 and FEV_1_ was found. These data highlight the potential for these AMPs to identify patients at greater risk of lung function decline. However, non-significant correlations of elafin, hBD-1, and SLPI with NTHI presence in the airways were found [308], indicating that the microbial type (viruses vs. bacteria) infecting COPD airways is a modifier of the AMP response.

**Table 3 ijms-25-01933-t003:** Mucus/sputum biomarkers in COPD.

Patient Cohorts	Sputum Samples	Outcomes	Reference
1 male COPD patient (diagnosed according to GOLD guideline 2017)	Spontaneous sputum	In the study, a new technology was used to reduce the complexity of clinical samples so as to optimize MALDI-TOF peptidome profiling.	Terracciano et al., 2019 [289]
90 COPD patients: 50 with a history of smoking (COPD/tobacco) and 40 who previously had TB (COPD/post-TB)	Spontaneous sputum	IL-1α, IL-1β, MIP-1β, sCD40L, and VEGF levels were higher in COPD patients compared to controls; IL-1α, IL-6, TNF-α, and IL-8 levels were higher in the COPD/tobacco patients compared to the COPD/post-TB patients.	Guiedem et al., 2020 [303]
31 patients with different diseases	Spontaneous/ induced sputum	Identification of proteins with potential applicability in clinical practice, e.g., markers of exacerbation.	Dasgupta et al., 2021 [290]
COPD patients (n = 14) with mean age of 65 years; asthmatic patients (n = 21) with mean age of 55	Induced sputum	Considering all patients, SLPI and hBD-1 were negatively correlated with ppFEV_1_ (*p* < 0.001, r = −0.610). SLPI and hBD-1 were higher in the COPD group compared to the asthma group, while elafin levels were not different.	Cane et al., 2021 [308]
Clinically stable COPD patients (n = 25) and COPD patients with exacerbations (n = 31)	Spontaneous sputum	The sputum cytokine signature of exacerbated patients differs from that of stable COPD patients. The observation that the levels of most cytokines do not stabilize with acute treatment of exacerbated patients suggests a prolonged effect of exacerbation on the status of COPD patients.	Barta et al., 2022 [291]
341 smokers with preserved spirometry, and 562 patients with COPD (178 with GOLD stage 1 disease, 303 with GOLD stage 2 disease, and 81 with GOLD stage 3 disease)	Induced sputum	The study identified several physiologic pathways altered in the airways of patients with COPD and associated with markers of disease severity, with the strongest relationships to metabolite biomarkers of mucus hydration and adenosine metabolism.	Esther et al., 2022 [294]
Cohort A: COPD patients (n = 30), healthy smokers (n = 10), and healthy non-smokers (n = 10); Cohort B: COPD patients (n = 81), healthy smokers (n = 15), and healthy non-smokers (n = 26)	Spontaneous sputum	Validated immunoassays applied to sputum supernatants demonstrated differences between COPD patients and controls, the effects of current smoking, and associations between *H. influenzae* colonization and higher levels of selected cytokines.	Mulvanny et al., 2022 [301]

GOLD: Global Initiative for Chronic Obstructive Lung Disease; hBD-1: human β-defensin-1; IL: interleukin; MIP: macrophage inflammatory protein; sCD40L: soluble cluster differentiation 40 ligand; SLPI: secretory leukoprotease inhibitor; TB: tuberculosis; VEGF: vascular endothelial growth factor.

### 4.3. Asthma (Table 4)

The use of sputum from asthmatic patients to detect inflammatory mediators, such as cytokines, represents an advantageous approach because these measurements are non-invasive, technically facile, and relatively low-cost. Second, cytokines are fundamental mediators of asthma pathogenesis, and their characterization can give new insights into immunologic mechanisms of this disease [309,310,311].

Schofield et al. [312] carried out an important study focused on the improvement of patient stratification through the identification of molecular sub-phenotypes of asthma defined by the proteomic signature. In their research, the proteomes of sputum from asthmatic patients were analyzed using an unbiased, label-free, quantitative mass spectrometry technique combined with topological data analysis. On the basis of proteomic profiles comparison, it has been possible to stratify patients in ten clusters representing three sub-phenotypes of asthma: highly eosinophilic, highly neutrophilic, and highly atopic with relatively low granulocytic inflammation. Interestingly, the insight into the type of granulocytic inflammation provided by these data could be useful for detection of targets for novel therapies.

Nevertheless, there is room for improvement in our accuracy in predicting treatment responses and a need for a better understanding of the underlying mechanisms. In a pioneering study, Gibson et al. [313] carried out a randomized, double-blinded, placebo-controlled trial (AMAZES) showing for the first time that patients with persistent symptomatic asthma could obtain beneficial effects from treatment with oral azithromycin for 48 weeks. Recently, Niessen et al. [314] investigated whether this macrolide exerts its beneficial effect by acting on TNF-α pathways, which are involved in neutrophilic asthma [315,316]. These researchers determined the concentrations, in sputum or serum, of soluble TNF receptor 1 (TNFR1), TNF receptor 2 (TNFR2), and TNF before and after 48 weeks of treatment with the antibiotic or placebo in a sub-cohort of the AMAZES study. At baseline, both TNFR1 and TNFR2 levels in sputum were significantly higher in neutrophilic asthma compared with non-neutrophilic asthma phenotypes, while their serum concentrations did not differ. Moreover, TNFR1 and TNFR2 levels in sputum correlated with decreasing pre-bronchodilator ppFEV_1_ and an increasing sputum neutrophils count. It is interesting to observe that serum TNFR1 was also increased in severe asthma, while sputum and serum TNFR2 were increased in patients who experienced frequent exacerbations. After treatment with azithromycin, the sputum TNFR2 and TNF were significantly lower compared to placebo, in particular in non-eosinophilic patients. The study [314] demonstrated that TNF markers, particularly their sputum levels, are strongly correlated to clinically important phenotypes of asthma, including neutrophilic and severe asthma. Noticeably, azithromycin can suppress the dysregulated TNF signaling and represents a potential approach for the treatment of asthma. It is clear that biomedical research is making great efforts to identify new biomarkers to improve the diagnosis and management of asthma.

Recently, Sato et al. [317] determined the concentration of neurturin (a neurotrophic factor that is fundamental for neuronal homeostasis) in asthma patients’ sputum. The role of this mediator in asthma is not clear, but some studies on animal models support the hypothesis that it could act as an anti-inflammatory molecule in allergic inflammatory processes [318,319]. The concentration of neurturin in sputum samples was detected by ELISA, and its serical concentrations did not correlate with its sputum levels. Interestingly, the sputum concentrations of neurturin were significantly higher in the atopic asthmatic subjects compared with non-atopic asthmatic subjects. The relationship between sputum neurturin and asthma symptoms, asthma severity, fixed airway obstructions, and asthma exacerbations is not clear. However, the levels of sputum neurturin were strongly correlated with fractional exhaled nitric oxide (FeNO), sputum eosinophils, IL-5, and IL-13 in induced sputum supernatant and negatively correlated with neutrophils and MMP9. These results would indicate that the sputum neurturin level is associated with Type 2 airway inflammation in adult asthmatic subjects.

One of the most challenging aspects in asthma management is represented by patients who are refractory to therapy. In this context, new treatments, such as the use of monoclonal antibodies, have been developed with the purpose of reducing the use of oral corticosteroids, which are responsible for long-term and costly side effects. For example, monoclonal antibodies targeting IL-5 or its receptor, IL-5R, are used for the treatment of severe eosinophilic asthma, with clinical improvements [320,321]. However, it seems that the response of patients to these biotherapies is somehow variable. In a study published in 2023, Moermans et al. [322] evaluated if there were biomarkers that can be used to determine the remission state in patients with severe eosinophilic asthma and treated with anti-IL-5 therapy. In the study, remission after 1 year of therapy was defined by evaluating the sputum level of some inflammatory mediators, such as eosinophil peroxidase (EPX), IgE, IL-3, IL-4, IL-5, IL-13, IL-25, IL-33, granulocyte macrophage colony-stimulating factor, thymic stromal lymphopoietin (TSLP), and eotaxin-1. Interestingly, in patients who experienced remission (11 of 52 enrolled), the sputum eotaxin-1, TSLP, IL-5, EPX, and IgE protein levels were increased at baseline and decreased after anti-IL-5 therapy.

The use of proteomics technology can help researchers to better delineate the different subtypes of asthma patients. Currently, asthma patients can be categorized into either ‘T helper 2-high’ (T2) or ‘non-T2’ disease (called endotypes) [323]. Recently, Gautam et al. [324] used an “omics technology” to identify sputum cytokine profiles that could help to better characterize the different disease subtypes so as to improve therapy in asthma patients. The study was based on a cross-sectional analysis of clinical features and sputum from 200 asthmatic patients and evaluated the profile of 10 cytokines belonging to alarmin, T2, and non-T2 pathways. On the basis of their results, these researchers identified three discrete signaling modules in patients with asthma. Surprisingly, the inclusion of alarmins allows for the separation of the canonical T2 cytokines into two unique modules: IL-5 grouped with TSLP alarmin and IL-13 grouped with IL-33. Moreover, patient clustering allowed for individuating two distinct endotypes associated with T2 immune signaling. It is noteworthy to specify that these results evidenced a new layer of immunologic heterogeneity within the T2 paradigm and suggest that the characterization of sputum in terms of cytokine proteomics may improve diagnosis as well as therapy in asthma patients.

To test the hypothesis that sputum inflammatory mediator profiles change between stable disease and exacerbation events, Ghrebe and colleagues [325] undertook a 1-year prospective study of moderate-to-severe adult asthmatics in stable and exacerbated conditions. Sputum pro-inflammatory and Type 1 (T1) immune mediators (IL-1β, IL-2, IL-6, IL-6R, IL-18, CXCL9, CXCL10, CCL5, TNFα, TNF-R1, TNF-R2, and chymotrypsin) were significantly increased at exacerbations compared with the stable state. The ROC area under the curves revealed that TNF-R2 and IL-6R were the strongest discriminators of an exacerbation, a finding that was validated in a pediatric group.

The mechanisms and immunological risk factors of severe pediatric asthma attacks, eventually triggered by viral respiratory tract infections, especially rhinovirus type C, are poorly understood. A real-world prospective observational study in children with acute and stable asthma was conducted to understand which sputum mediators are associated with virus-triggered asthma attacks [326]. In acute asthma, characterized by increased neutrophils counts in sputum, there was a parallel rise of acute-phase biomarkers and neutrophil attractants, including IL-6 and IL-8; cytokines associated with bacterial signals, such as TNF-R1 and TNF-R2; IL-5, a typical type-2 inflammation; and T-cell attractant cytokines, associated with viral infections, such as CCL-5, CXCL-10, CXCL-11, and CXCL-9. Increased MUC5AC levels, and a decreased MUC5B:MUC5AC ratio, were present in the sputum of children with acute asthma, potentially altering mucus composition and contributing to the airway mucus obstruction observed during acute asthma [36,47,48,49].

The contribution of natural AMPs to asthma has been investigated, being both pro-inflammatory (LL-37 and hBD-1) and protective (hBD-2) [327]. A recent article showed that asthmatic patients display SLPI and hBD-1 in their sputum; however, these AMP levels were higher in the healthy controls and COPD group compared to the asthma group, while elafin did not differ among groups [308]. Moreover, no differences in AMP levels were noted between low and high neutrophil burdens, confirming a previous study showing a lack of correlation of sputum hBD-1 and inflammatory groups [229]. Further studies with higher numbers of patients are warranted to better comprehend whether AMPs may be prognostic and stratification biomarkers in asthma.

**Table 4 ijms-25-01933-t004:** Mucus/sputum biomarkers in asthma.

Patient Cohorts	Sputum Samples	Outcomes	Reference
246 patients: 118 nonsmoking patients with severe asthma; 48 current or ex-smoking patients with severe asthma; 40 patients with mild-to-moderate asthma	Induced sputum	The current classifications of asthma based on cell count (eosinophils and neutrophils) were improved through proteomic characterization.	Schofield et al., 2019 [312]
102 adults and 34 children with moderate-to-severe asthma	Spontaneous or induced sputum	TNF-rcceptor2 and IL-6 receptor were the strongest discriminators of an exacerbation in both adults and children.	Ghebre et al., 2019 [325]
AMAZES clinical trial (ACTRN12609000197235)	Spontaneous sputum	TNF sputum marker activity correlates with asthma severity and decreases after azithromycin treatment.	Niessen et al., 2021 [314]
48 children (acute asthma, n = 18; stable asthma, n = 17)	Induced sputum	CXCL-10, CCL-5, and TNF-R2 were the strongest discriminators of an asthma attack.	Ramphul et al., 2021 [326]
Asthmatic patients (n = 21) with mean age of 55; COPD patients (n = 14) with mean age of 65 years	Induced sputum	Considering all patients, SLPI and hBD-1 were negatively correlated with ppFEV_1_. SLPI and hBD-1 were lower in the asthma group compared to the COPD group, while elafin levels were not different.	Cane et al., 2021 [308]
65 adult asthmatic subjects	Induced sputum	Sputum neurturin is a new marker for type 2 airway inflammation.	Sato et al., 2023 [317]
52 patients with severe asthma (51 patients treated with mepolizumab and one patient treated with reslizumab)	Induced sputum	Baseline type 2 airway inflammation markers can predict remission in severe eosinophilic asthma treated with anti-IL-5 agents.	Moermans et al., 2023 [322]
200 asthmatic patients	Induced sputum	Three groups of highly correlated cytokines and alarmins were identified: two T2 modules, the TSLP_mod_ (TSLP, IL-4, IL-5, IL-9) and the IL-33_mod_ (IL-33, IL-13, IL-21), and one non-T2 module, the IL-1β_mod_ (IL-1β, IL-6, G-CSF). The TSLP_mod_ was associated with asthma severity, airway obstruction, eosinophilia, and elevated FeNO.	Gautam et al., 2023 [324]
65 adult asthmatic subjects	Induced sputum	Sputum neurturin is a new marker for type 2 airway inflammation.	Sato et al., 2023 [317]

AMAZES: Asthma and Macrolides: The Azithromycin Efficacy and Safety study; FeNO: fractional exhaled nitric oxide; G-CSF: granulocyte colony stimulating factor; hBD-1: human β-defensin-1; IL: interleukin; SLPI: secretory leukoprotease inhibitor; TNF: tumor necrosis factor; TSLP: thymic stromal lymphopoietin.

## 5. Concluding Remarks

Measurements of mucus properties through micro-rheology, low-field NMR, osmotic pressure, and so on have highlighted the presence of a link between the structure of mucus and its macroscopic and mesoscopic viscoelastic properties. Such a complex relationship originates from the variety of molecules involved, mainly mucins and proteins, and from the different interactions between them. These interactions are currently the subject of thorough studies, both for healthy and CRD patients. Interestingly, the combined use of *LF-NMR* and rheology (both at the macro and the micro scale) represents a powerful tool for the characterization of the mucus nanostructure (such as mesh size distribution). Indeed, rheology can provide average information on the crosslinks network elastically active and on the solid content by means of the elastic (*G*′) and viscous (*G*″) moduli, respectively. On the other side, *LF-NMR*, essentially depending on the solid surface/water volume ratio inside mucus, allows for determining the mesh size distribution. In addition, it provides a simple and reliable parameter (spin–spin or transverse relaxation time *T*_2_) that correlates with many biomarkers of the patient’s clinical condition. An in-depth understanding of mucus structure would form the basis for targeted symptomatic therapy and the development of ad hoc engineered drug delivery systems. Indeed, from the structural point of view, the pathological aspects of mucus consist of both the increase in solids content (mainly mucins and other proteins) and the formation of a highly interconnected, three-dimensional network. This network, in turn, hinders the in situ delivery of typical administered drugs (antibiotic, anti-inflammatory, and mucolytic) that must cross the small network meshes in order to be therapeutically active. Moreover, the formation of a three-dimensional network increases mucus’s viscous and elastic properties. This, in turn, makes ineffective the mucociliary clearance as cilia cannot transport away the highly viscoelastic fluid that, ultimately, becomes a static environment, which is the ideal nest for pathogen (typically bacteria) growth.

The discovery and implementation of noninvasive, sensitive, specific, and predictive biomarkers represent the holy grail of human pathologies, including CRDs. In asthma, COPD, and CF, some studies have identified biomarkers for which the levels differ from those in normal subjects and that can predict acute exacerbation events, though still lacking in a robust appraisal of their modification upon old and novel treatments. This lack of knowledge concerning these aspects also involves the mucus structure and viscoelasticity features, which are important in mucociliary clearance and pathophysiology. As pulmonary functional tests, such as ppFEV_1_, are still the gold standard for diagnostic, prognostic, and therapeutic response purposes, a wealth of work is still needed for embracing these measures as correlates of disease severity in association with (or even as substitutes of) pulmonary functional tests.

## Figures and Tables

**Figure 1 ijms-25-01933-f001:**
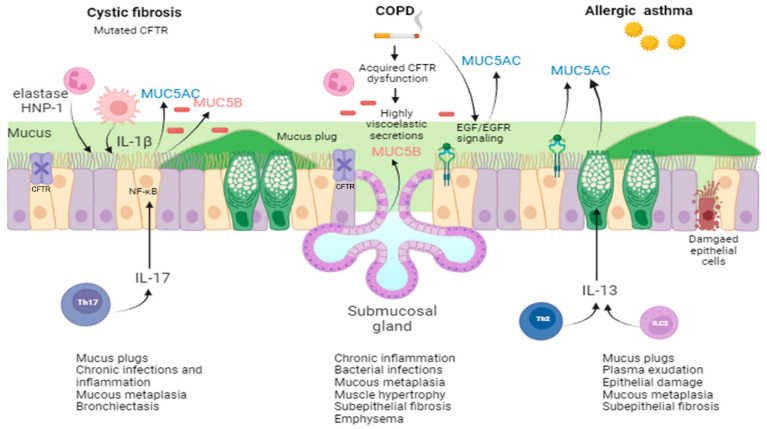
Mediators of mucin expression and histopathologic changes in CRDs. CF is caused by CFTR mutations (with a cross on the CFTR molecule) instigating a IL-17-driven neutrophil chronic inflammation. CFTR dysfunction has also been found to be associated with COPD, in which mucus overproduction is also elicited by EGFR signaling. Allergic asthma is marked by IL-13 and EGFR-mediated activation of mucin expression and secretion. See text for further details. CFTR: Cystic Fibrosis Transmembrane Conductance Regulator; EGF: Epidermal Growth Factor; EGFR: EGF Receptor; HNP-1: Human Neutrophil Peptide-1.

**Figure 2 ijms-25-01933-f002:**
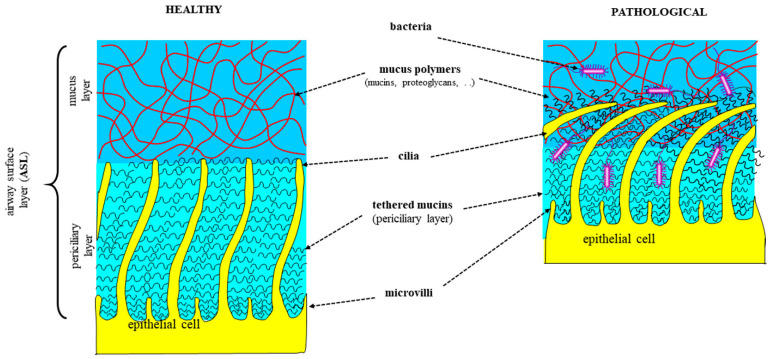
Schematic representation of the airway surface layer (ASL) in normal (healthy) and pathological conditions. In order to improve clarity, the thicknesses of the periciliary and the mucus layers are not in scale. Adapted from Ref. [145].

**Figure 3 ijms-25-01933-f003:**
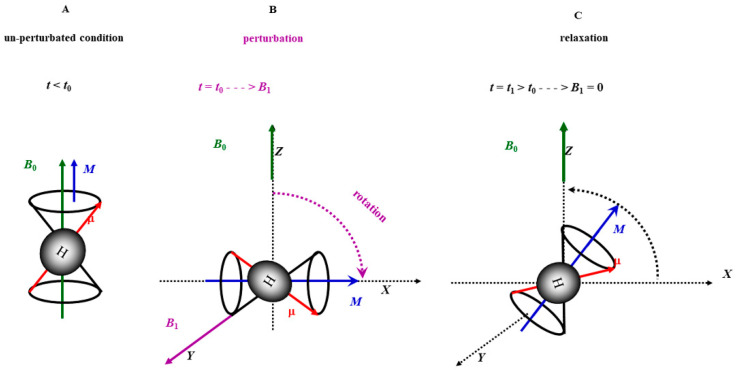
Working principle of Low-Field NMR (LF-NMR). (**A**) In the presence of an external and constant magnetic field (0.37 T ≤ *B*_0_ ≤ 2.43 T), the permanent dipole (μ–red arrow) of hydrogen atoms tends to align with the *B*_0_ direction (conventionally, the Z-direction; *t* < *t*_0_). More precisely, every μ starts rotating at the so-called Larmor frequency around *B*_0_ (green arrow), forming a characteristic angle with *B*_0_ (Z-direction). The vector sum of μ competing with all of the hydrogen atoms gives origin to the induced magnetization vector (M–blue arrow) that is parallel to *B*_0_. (**B**) The application of a perturbation (a radio frequency pulse *B*_1_ (violet arrow) perpendicular to *B*_0_ and rotating in the XY plane at the Larmor frequency) provokes the progressive M rotation in the X-Y plane (*t*_0_ < *t* ≤ *t*_1_). (**C**) After *B*_1_ removal (*t* > *t*_1_), M comes back to the original *B*_0_ direction (relaxation). The relaxation process, implying the disappearance of the M projection in the X-Y plane, is characterized by the so-called spin–spin or transverse relaxation time *T*_2_. Interestingly, *T*_2_ depends on the *B*_0_ intensity, the temperature, and the chemical, physical, and topological characteristics of the environment embedding the water molecules carrying the relaxing hydrogen atoms [159].

**Figure 4 ijms-25-01933-f004:**
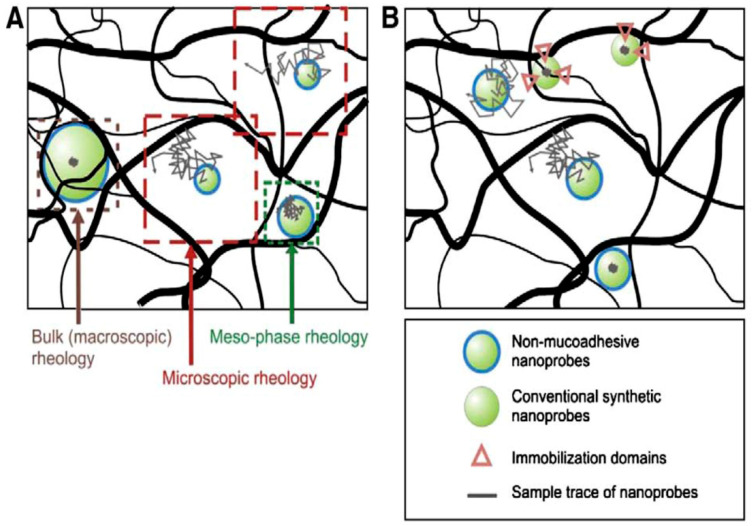
(**A**) Schematic representation of the length scale dependence of viscosity in a nanoscopically heterogeneous fluid. Non-adhesive particles that are significantly smaller than the mesh spacing undergo Brownian diffusion and probe the microscopic rheology. As the particle size approaches the dimensions of the mesh spacing, particle movement becomes hindered by the mesh microstructure at short time scales, leading to a mesophase rheology regime. Particles that are significantly larger than the mesh spacing probe the bulk or macroscopic rheology of the gel. (**B**) Schematic comparison of non-mucoadhesive rheological nanoprobes and conventional polymeric particles. Conventional nanoprobes are immobilized to mucin fibers via adhesive interactions. Their strongly hindered motion, as reflected by the small dimensions of the traces, suggests a markedly higher viscoelastic environment than the true local viscoelasticity of mucus. In contrast, the motion of non-mucoadhesive nanoprobes correctly reflects the local viscous and elastic contributions from the mucus mesh architecture. Reprinted from Ref. [11] Advanced Drug Delivery Reviews, Vol. 61, Samuel K. Lai, Ying-Ying Wang, Denis Wirtz, Justin Hanes, Micro- and macrorheology of mucus, pages 86–100, 2009, with permission from Elsevier.

**Figure 5 ijms-25-01933-f005:**
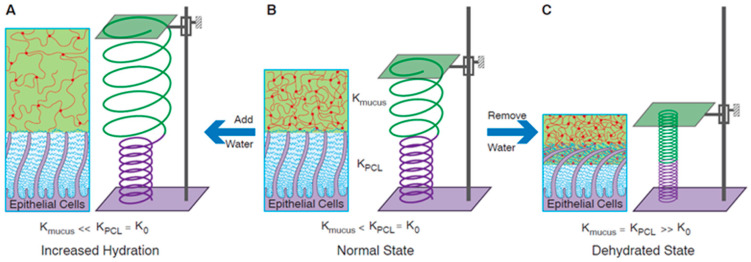
(**A**–**C**) Schematic illustrations showing the effects of the relative water-drawing powers of the mucus gel and the PCL. (**B**) Normal state: The osmotic modulus of normal mucus is smaller than that of the PCL, represented by a green spring (K_mucus_) with a diameter larger than a purple spring (K_PCL_ = K_0_). The volume of water in the system is depicted by the fixed distance between two plates. (**A**) Increased hydration: Water added to the healthy airway surface (distance between plates increased) with K_mucus_ < K_0_ preferentially enters and thus dilutes the mucus layer, leaving the PCL unchanged. The resulting osmotic modulus of the mucus layer is much smaller than that of the PCL (K_mucus_ << K_0_). This state is depicted by the increased length and diameter of the green spring, with no change in the purple spring. (**C**) Dehydrated state (plates close to each other): As water is removed, it first preferentially leaves the mucus gel because of its lower osmotic modulus. Further dehydration leads to removal of water from both the mucus gel and the PCL. The moduli of both layers are increased and equal, represented by smaller diameters of shortened springs. This state corresponds to diseased airways (COPD and CF). From Ref. [145] Science, Vol. 337, Brian Button, Li-Heng Cai, Camille Ehre, Mehmet Kesimer, et al., A Periciliary Brush Promotes the Lung Health by Separating the Mucus Layer from Airway Epithelia, pages 937–941, 2012. Reprinted with permission from AAAS.

**Table 1 ijms-25-01933-t001:** Pathological features of sputum production in CRDs.

Pathological Changes/Disease	COPD	Asthma	CF
Mucus amount	Increased luminal content	Increased luminal content	Increased luminal content
Mucins	- Increased amounts of MUC5AC and MUC5B - Small amount of MUC2	- Increased content of MUC5AC and variable content of MUC5B - Small amount of MUC2	Spontaneous or induced sputum: - Decreased amounts of MUC5AC and MUC5B - Small amounts of MUC2 - Increased levels of MUC5AC and MUC5B BAL: - Increased amounts of MUC5B and MUC5AC
Other components of mucus	Albumin	DNA, actin	DNA, actin
“Tethering” of mucus to goblet cells	No	Yes	No
Mucus viscosity	High	High	Low
Airway pathology	- Goblet cell hyperplasia - Submucosal gland hypertrophy	- Loss of ciliated cells - Marked goblet cell hyperplasia - Submucosal gland hypertrophy	- Goblet cell hyperplasia - Submucosal gland hypertrophy
Plasma exudation	No	Yes	No
Airway inflammation	Macrophages and neutrophils	- Eosinophils and Th2 cells - Neutrophils in severe disease	Macrophages and neutrophils

## Data Availability

Not applicable.

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
