# Peer review of "Mucus Structure, Viscoelastic Properties, and Composition in Chronic Respiratory Diseases"

_ijms, 2024, doi:10.3390/ijms25031933_

Round 1

Reviewer 1 Report

Comments and Suggestions for Authors

The presented manuscript is a very important and timely review. The problem of mucus formation and its diagnostic role in many diseases (COPD, asthma, cystic fibrosis, etc.) are either poorly described or destructured in different articles. The presented manuscript is well written, conveniently structured, and includes a large number of modern literature sources on the issue.

At the same time, the authors do not clearly indicate the criteria for inclusion and exclusion of articles that formed the basis of this review. I recommend indicating this, for example, in the final paragraph of the introduction (lines 60–66).

Also, the quality of Figures 1, 4 and 5 should be increased. There is a feeling that they were copied from other sources (Although it is indicated that figure 1 was drawn in Biorender.com.). In this case, they should be excluded or another version of the figure should be presented.

Author Response

The presented manuscript is a very important and timely review. The problem of mucus formation and its diagnostic role in many diseases (COPD, asthma, cystic fibrosis, etc.) are either poorly described or destructured in different articles. The presented manuscript is well written, conveniently structured, and includes a large number of modern literature sources on the issue.

At the same time, the authors do not clearly indicate the criteria for inclusion and exclusion of articles that formed the basis of this review. I recommend indicating this, for example, in the final paragraph of the introduction (lines 60–66).

R1: We thank the Reviewer for appreciation of our effort in presenting a timely review on respiratory mucus formation and composition in health and diseased states. According to her/his request we have added at the end of the Introduction the following paragraph: “This paper was conceived as a narrative review and not a systematic one, thus we have surveyed the most significant and important papers in the fields of healthy and pathological mucus and CRDs by performing an electronic search in the PubMed, EMBASE, and Scopus databases.”

Also, the quality of Figures 1, 4 and 5 should be increased. There is a feeling that they were copied from other sources (Although it is indicated that figure 1 was drawn in Biorender.com.). In this case, they should be excluded or another version of the figure should be presented.

R2: Higher resolution (300 dpi) TIFF images have been provided for all the figures presented in the review.

Reviewer 2 Report

Comments and Suggestions for Authors

This is a comprehensive review of airway mucus biophysical properties in muco-obstructive disease. While I appreciate the authors' efforts, the reliance solely on sputum studies in humans undermines the significance of this manuscript due to the lack of attention to normal mucus. Below are my comments:

1.      The manuscript highlights how mucin concentration affects biophysical properties, but the role of ions, particularly pH as shown in CF studies, is another crucial factor. Adding a paragraph on this topic would be valuable.

2.      While sputum studies are insightful, incorporating insights from healthy mucus, like those from CF animal models (especially pigs), would provide a stronger foundation for understanding disease states. This could be covered before discussing advanced diseases in sputum.

3.      Current consideration of MUC5AC/B overexpression contributing to mucus hypersecretion, which misses other contributors like goblet cell hyperplasia and submucosal gland hypertrophy. Broadening the discussion of hypersecretion causes to include these factors would be beneficial.

4.      Including a paragraph on glandular mucus generation by submucosal glands, its differences from goblet cell mucus, and recent research in this area would enrich the manuscript and address current efforts to understand these distinctions.

5.      Antimicrobials should be considered as important elements for mucus function. In section 4, the authors focused on sputum biomarkers for disease. However, the authors should introduce key antimicrobial proteins in mucus and their function before going to airway disease.

6.      Cell and mucus orientation should be flipped in Figure 2.

Author Response

This is a comprehensive review of airway mucus biophysical properties in muco-obstructive disease. While I appreciate the authors' efforts, the reliance solely on sputum studies in humans undermines the significance of this manuscript due to the lack of attention to normal mucus. Below are my comments:

  1. The manuscript highlights how mucin concentration affects biophysical properties, but the role of ions, particularly pH as shown in CF studies, is another crucial factor. Adding a paragraph on this topic would be valuable.

R1: The authors thank the Reviewer for the kind appreciation of our efforts, providing a revised version of the review with novel paragraphs and sections addressing her/his comments.

Thus, to comply with the topic of ions and pH affecting biophysical properties of mucus, we have added a novel subsection within the Section 3 and named “3.3. The role of pH and ionic strength on mucus properties”.

  1. While sputum studies are insightful, incorporating insights from healthy mucus, like those from CF animal models (especially pigs), would provide a stronger foundation for understanding disease states. This could be covered before discussing advanced diseases in sputum.

R2: We have now included a novel Subsection, called “1.1. Respiratory mucus in healthy and diseased conditions”, which described main features of healthy mucus. Also references to mucus in animal models (especially CF pig and rat models) are given herein and in other Sections (1.3.1. Physiology of respiratory mucus secretion; 1.3.2. Mucins, goblet cells and submucosal glands in pathophysiology of mucus secre-tion; 1.4. Pathophysiology of mucus production in CRDs; 3.3 The role of pH and ionic strength on mucus properties) .

  1. Current consideration of MUC5AC/B overexpression contributing to mucus hypersecretion, which misses other contributors like goblet cell hyperplasia and submucosal gland hypertrophy. Broadening the discussion of hypersecretion causes to include these factors would be beneficial.

R3: We have now included within the Subsection “1.3.2. Mucins, goblet cells and submucosal glands in pathophysiology of mucus secretion” a discussion on the contribution of globet cell hyperplasia/metaplasia and submucosal galnd hypertrophy to hypersecretion in CF, COPD, and asthma.

  1. Including a paragraph on glandular mucus generation by submucosal glands, its differences from goblet cell mucus, and recent research in this area would enrich the manuscript and address current efforts to understand these distinctions.

R4: We have now added a novel Subsection, named “1.3.1. Physiology of respiratory mucus secretion” about these topics.

  1. Antimicrobials should be considered as important elements for mucus function. In section 4, the authors focused on sputum biomarkers for disease. However, the authors should introduce key antimicrobial proteins in mucus and their function before going to airway disease.

R5: We agree with the Reviewer that the antimicrobial topic in mucus/sputum was underrepresented in our review. Thus, we have now included a new paragraph within Section 4 presenting and discussing the presence and role of antimicrobial substances in mucus/sputum in CF, COPD, and asthma. Moreover, we refer to the role of antimicrobial proteins and peptides in CRDs in other places of the review (1.3.1. Physiology of respiratory mucus secretion ; 1.3.2. Mucins, goblet cells and submucosal glands in pathophysiology of mucus secretion; 1.4. Pathophysiology of mucus production in CRDs).

  1. Cell and mucus orientation should be flipped in Figure 2.

R6: Fig. 2 was flipped.

Round 2

Reviewer 2 Report

Comments and Suggestions for Authors

The authors have addressed all my comments.